# PathChat-SegR1: Reasoning Segmentation in Pathology via SO-GRPO

**Zelin Liu**[1][*]**, Dongdong Chen**[1][*]**, Yusong Sun**[1]**, Yuqi Hu**[1]**, Huang Jie**[2]**,**
**Sicheng Dong**[1]**, Xu Han**[1]**, Hongmei Yi**[2]**, Qiyuan Bao**[2][†]**, Lichi Zhang**[1][†]
[1]Shanghai Jiao Tong University    [2]Shanghai Ruijin Hospital

## Abstract

Segmentation in pathology image requires handling out-of-domain tissue morphologies and new pathologies beyond training distributions, where traditional closed-set segmentation approaches fail to generalize. Reasoning segmentation enables zero-shot generalization via prompting with text queries. However, existing reasoning segmentation models face three barriers when applied to pathology: (1) the vision encoder lack pathology-specific knowledge and robustness to staining variations, (2) the large language model (LLM) backbone for reasoning fails to identify whether it has gathered sufficient semantic context to trigger the segmentation output, and (3) no reasoning segmentation benchmarks and datasets exist for pathology analysis. Consequently, we introduce PathChat-SegR1, a reasoning segmentation model built upon pathology-specific vision encoders trained with a novel stain-invariant self-distillation for robust pathology image representations. Moreover, we propose Segmentation-Optimized GRPO (SO-GRPO), a reinforcement learning method specifically for reasoning segmentation that learns to determine optimal segmentation timing based on accumulated reasoning context. Finally, we construct a pathology-specific reasoning segmentation benchmark of 118,667 triplets of pathology image, ground-truth mask, query, and reasoning chain including both public and private pathology images. Zero-shot evaluation on pathology images with out-of-domain morphologies/pathologies shows 61% improvement over state-of-the-art segmentation models. More details: https://github.com/yul945562-bit/Pathseg.

## 1 Introduction

Segmentation on pathology images enables quantitative tissue analysis to support clinical diagnosis (Zhu et al., 2025b; Wang et al., 2019; Zhou et al., 2024b). However, clinical practice routinely encounters rare pathologies or new morphologies absent from training sets (Schäfer et al., 2024; Zhou et al., 2024a), where closed-set segmentation methods fail to generalize. When pathologists examine uncommon tumor pathologies/morphologies, they typically apply domain knowledge and clinical terminology to identify malignant regions, which is a reasoning process that existing closed-set segmentation models cannot also replicate. Although SAM (Kirillov et al., 2023; He et al., 2025) and variants offer open-set segmentation capabilities through prompting (*e.g.*, bounding boxes or point coordinates), they require intensive-manual clicking for every target instance. Moreover, they cannot interpret semantic or clinical queries expressed in text. Finally, beyond generalization to new pathologies/morphologies, clinical pathology workflows also require explainable segmentation models, which can articulate diagnostic reasoning rather than making predictions solely (van Veldhuizen et al., 2025; Wu et al., 2023).

Reasoning segmentation offers a promising pathway for open-set segmentation in pathology images. Specifically, reasoning segmentation takes queries encoded in natural language to guide the segmentation on unseen objects via semantic understanding and reasoning (Lai et al., 2024). Moreover, reasoning segmentation models may generate explicit reasoning chains, such that pathologists can validate against clinical knowledge (Moor et al., 2023; Lin et al., 2025). Formally, Table 1

---

[*]Equal contribution.
[†]Corresponding authors: bqy12145@rjh.com.cn, lichizhang@sjtu.edu.cn.

Table 1: Comparison of segmentation methods for pathology. Abbreviations: "Unseen Gen.": Generalization to unseen morphologies/objects; "Path. Spec.": Pathology-specific models; "Reason.": Capability for general visual reasoning and language understanding; "Stain Rob." for stain-variation-invariant representations; "RL-Seg" for segmentation-specific reinforcement learning.

| Method | Unseen Gen. | Path. Spec. | Reason. | Stain Rob. | RL-Seg | Interaction |
|---|---|---|---|---|---|---|
| nnU-Net (Isensee et al., 2021) | ✗ | ✗ | ✗ | ✗ | ✗ | ✗ |
| MedSAM (Ma et al., 2024) | ✓ | ✗ | ✗ | ✗ | ✗ | Point/Box |
| SAM-Path (Zhang et al., 2023) | ✓ | ✓ | ✗ | ✗ | ✗ | Point/Box |
| SegAnyPath (Wang et al., 2024c) | ✓ | ✓ | ✗ | ✓ | ✗ | Point/Box |
| BiomedParse (Zhao et al., 2024) | ✓ | ✗ | ✗ | ✗ | ✗ | Text |
| Seg-Zero (Liu et al., 2025b) | ✓ | ✗ | ✓ | ✗ | ✗ | Text |
| LISA (Lai et al., 2024) | ✓ | ✗ | ✓ | ✗ | ✗ | Text |
| MMR (Jang et al., 2025) | ✓ | ✗ | ✓ | ✗ | ✗ | Text |
| **PathChat-SegR1 (Ours)** | ✓ | ✓ | ✓ | ✓ | ✓ | Text |

compares reasoning segmentation methods with existing closed-set segmentation methods for pathology analysis. However, existing reasoning segmentation models rely on general-domain vision encoder that lack specialized understanding of pathological features, as well as the robustness to stain variations (Wang et al., 2024b; Ke et al., 2025). Consequently, these models exhibit suboptimal generalization ability when applied to new pathologies/morphologies. Moreover, LISA's paradigm of inserting `<SEG>` tokens at predetermined positions prevents autonomous determination of optimal segmentation timing. Additionally, although reinforcement learning methods such as GRPO (Shao et al., 2024) can train reasoning segmentation models to generate reasoning chains and segmentation accuracy concurrently (Liu et al., 2025b), they fail to identify when the reasoning segmentation model has accumulated sufficient semantic context to trigger segmentation output. This also results in suboptimal performance in pathology segmentation, which requires more complex reasoning. Finally, there is no existing reasoning segmentation dataset or benchmarks for pathology.

To address these limitations, we introduce PathChat-SegR1, a pathology-specific reasoning segmentation model. Specifically, PathChat-SegR1 builds upon the pathology-specific vision encoder with novel reinforcement learning designed for segmentation. We reformulate `<SEG>` token emission from fixed insertion to autonomous generation. To further improve performance, we also train the visual encoder in PathChat-SegR1 with stain-invariant self-distillation to handle stain variations.

In brief, the major contributions of this paper are three-fold. First, we propose PathChat-SegR1, a reasoning segmentation model for pathology with a pathology-specific vision encoder fine-tuned with a novel stain-invariant self-distillation. Second, we propose Segmentation-Optimized GRPO (SO-GRPO), which extends standard GRPO specifically for reasoning segmentation by identifying the optimal `<SEG>` generation timing during LLM reasoning. Specifically, we introduce a differentiable segmentation-performance reward, and sparsity-aware reward for controlling redundant generation, together with an adaptive scheduling for ensuring training stability. Finally, we construct a large-scale PathChat-SegR1 benchmark comprising 118,667 triplets of pathology image, ground-truth mask, query, and reasoning chain, spanning across both public and private pathology images. Correspondingly, we propose a novel semi-automated pipeline to annotate the reasoning chain.

## 2 RELATED WORKS

**Pathological Image Segmentation** Segmentation on pathology images helps quantify tissue morphology to support clinical diagnosis and treatment planning. Traditional closed-set segmentation approaches leverage general medical segmentation models (*e.g.*, nnU-Net, MedSAM, BiomedParse) trained on with pre-defined categories of pathology and limited morphology types. Yet, these methods lack pathology-specific understanding and therefore fail to generalize to new tumor-pathologies and morphologies absent from training distributions. SAM-based adaptations attempt to address generalization via box or point prompting (Kirillov et al., 2023; Shen et al., 2025a; Shi et al., 2025). For example, SAM-Path (Zhang et al., 2023) adapts SAM (Kirillov et al., 2023) via trainable class embeddings to represent different morphologies, but requires extensive data set-specific fine-tuning for each new pathology/morphology type, similarly to SegAnyPath (Wang et al., 2024c; Shen et al.,

2025d). However, prompting in SAM variants demands intensive manual efforts, rendering them impractical for whole slide images containing hundreds of cellular structures.

In summary, existing pathology segmentation methods achieve strong performance in training distributions under closed-set assumptions; or open-set capabilities at the cost of heavy manual prompting efforts, but not both simultaneously (Shen et al., 2025a; Shi et al., 2025).

**Reasoning Segmentation**    Reasoning segmentation generates pixel masks from text queries to avoid time-consuming point/box prompting (Lai et al., 2024; Shen & Zhang, 2025; Shen et al., 2025b). There are three major paradigms for building a reasoning segmentation model. The first paradigm employs multimodal LLMs (MLLMs) to generate prompts, *i.e.*, points or bounding boxes, to guide a separate frozen segmentation model For example, SAM4MLLM and Seg-Zero (Chen et al., 2024; Liu et al., 2025b) generates coordinates from MLLM's reasoning to prompt SAM (Kirillov et al., 2023) for subsequent segmentation.

However, these types of methods suffer from performance limitations and remain sensitive to prompts as the point prompting inadequately captures the complex spatial information in pathology images. The second paradigm integrates the segmentation capability directly into LLM by connecting the segmentation decoder to LLM with special tokens. For example, LISA (Lai et al., 2024) introduces the `<SEG>` token whose hidden representation feeds into a mask decoder to guide the mask segmentation, while MMR (Jang et al., 2025) extends this approach to enable multi-granularity reasoning for hierarchical tissue structures. The third paradigm reformulates segmentation as text generation (Shen et al., 2025c), with Text4Seg++ (Lan et al., 2025) labeling image patches using textual categories to enable a direct text-to-mask pathway. Despite these advances, existing reasoning segmentation models rely on a general-domain vision encoder that lacks specialized understanding of pathology features and robustness to staining variations, resulting in limited performance.

**Reinforcement Learning for Segmentation**    Reinforcement learning formulates segmentation as sequential decision-making to improve model performance Existing methods intervene at different stages, including pixel-level approaches (Liu et al., 2025a; Araslanov et al., 2019) treat each pixel as an agent with actor-critic policies but remain computationally costly for whole slide images, while auxiliary localization methods (Yi et al., 2023) train boundary-seeking controllers under PPO to reduce annotation requirements. Recent work focuses on prompting and reasoning optimization, where agents learn to produce informative prompts for foundation models. For example, Align-SAM (Huang et al., 2024) applies PPO for point-prompt selection, while Seg-Zero, MedReasoner, and LENS (Liu et al., 2025b; Yan et al., 2025; Zhu et al., 2025a) jointly optimize chain-of-thought reasoning and spatial prompts via GRPO. However, standard GRPO assigns rewards uniformly across complete generation trajectories, preventing segmentation models from identifying optimal timing for segmentation generation during the reasoning process.

## 3    METHODS

### 3.1    OVERVIEW OF PATHCHAT-SEGR1

Fig. 1 illustrates the architecture of PathChat-SegR1. Given a pathology image and text query, PathChat-SegR1 produces both a reasoning chain in text and a segmentation mask through its three components. First, the vision-language model (VLM) backbone leverages the RuiPath encoder to extract pathology-specific features that inform reasoning chain generation, which contains a `<SEG>` token to initiate segmentation. Secondly, the MedSAM encoder processes the pathology image separately to capture fine-grained spatial information needed for accurate mask generation. Finally, an adapter then bridges the `<SEG>` token representation from the VLM to guide the mask decoder in producing the final segmentation output. Training of PathChat-SegR1 goes through three stages. During pre-training, the VLM learns pathology-specific knowledge, where the MedSAM encoder undergoes stain-invariant self-distillation concurrently to handle staining variations. During supervised fine-tuning, RathChat-SegR1 learns to align vision and language embeddings for reasoning segmentation. To save training costs, this stage applies LoRA to the VLM, adapters to the vision encoders, and full-parameter training to the mask decoder. During SO-GRPO reinforcement learning, RathChat-SegR1 improves its reasoning ability by learning to determine when a sufficient semantic context has accumulated to generate the `<SEG>` token in the appropriate reasoning steps.

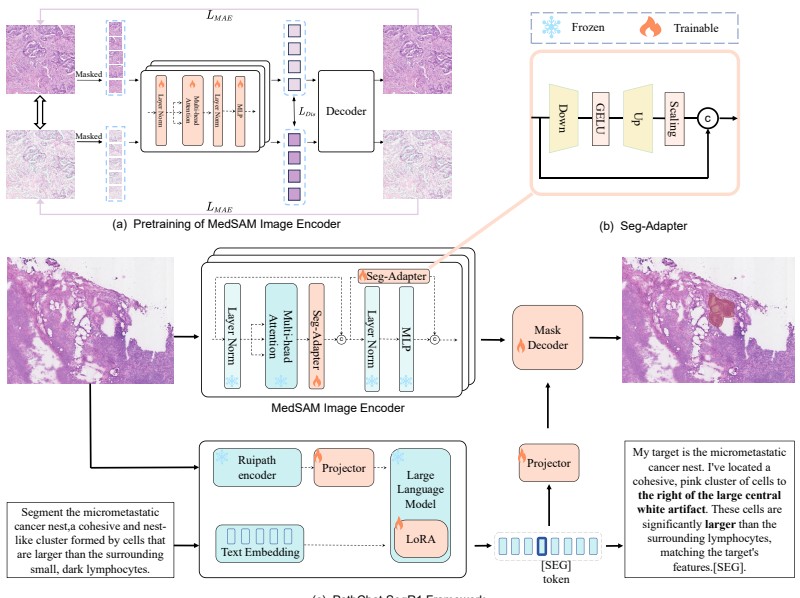

Figure 1: PathChat-SegR1 architecture overview.

## 3.2 STAIN-INVARIANT VISUAL PRE-TRAINING

Pathology images exhibit staining variations from different preparation protocols and laboratory practices, causing reasoning segmentation models trained on specific staining appearances to fail when reasoning about tissue morphology in images with different staining styles (Shen et al., 2022; Wang et al., 2024b; Ke et al., 2025). To address this, we apply stain-invariant self-distillation to the pre-training of MedSAM vision encoder through masked auto-encoding combined with LAB color space augmentation to learn features that remain consistent across stain variations. Specifically, for each pathology image $x_i$, we generate two virtually re-stained views (with $z_i^a$ and $z_i^b$ denoting its embedding) through random channel-wise linear transformations in the LAB color space via RandStainNA (Ke et al., 2023; Shen et al., 2022). The MedSAM vision encoder learns to reconstruct the masked patches from pathology images via masked autoencoding (He et al., 2021) while maintaining consistent feature representations across the two staining variations. Formally, the pre-training objective $\mathcal{L}_{\text{SSL}}$ of MedSAM vision encoder is formulated as:

$$\mathcal{L}_{\text{SSL}} = \frac{\alpha}{N} \sum_{i=1}^{N} \text{Cos}(\mathbf{s}_{i,1}, \mathbf{s}_{i,2}) \|z_i^a - z_i^b\|^2 + \mathcal{L}_{\text{MAE}} \tag{1}$$

where $\alpha$ is a balancing coefficent, $\mathbf{s}_{i,1} = [A_{v,1}, D_{v,1}]$ and $\mathbf{s}_{i,2} = [A_{v,2}, D_{v,2}]$ represent stain template vectors extracted from the two augmented views (Shen et al., 2022; Wang et al., 2024b), with $A$ and $D$ denoting the mean and STD of the color distribution in LAB space. The cosine distance term weights the feature discrepancy by the magnitude of the stain variation, assigning higher penalties when the staining differences are larger. The masked auto-encoding loss $\mathcal{L}_{\text{MAE}}$ enforces reconstruction of masked regions. Here, the MedSAM vision encoder serves to extract fine-grained spatial features needed for precise mask generation. MedSAM processes the pathology image independently from the VLM pathway, providing complementary spatial information that the adapter later combines with the <SEG> token representation.

Meanwhile, the VLM backbone of PathChat-SegR1 leverages Qwen2.5VL (Bai et al., 2025) and replaces the vision encoder with RuiPath, a foundation model pre-trained on large-scale histopathology data using DINOv2 (Oquab et al., 2024). RuiPath provides pathology-specific visual understanding, including tissue architecture, cellular morphology, and disease patterns absent from natural image datasets. The stain-invariant self-distillation described above also applies to the RuiPath vision encoder during the pre-training of VLM. During VLM pre-training, we also train the LLM on pathology image-text pairs through next-token prediction, where the LLM learns to generate textual descriptions

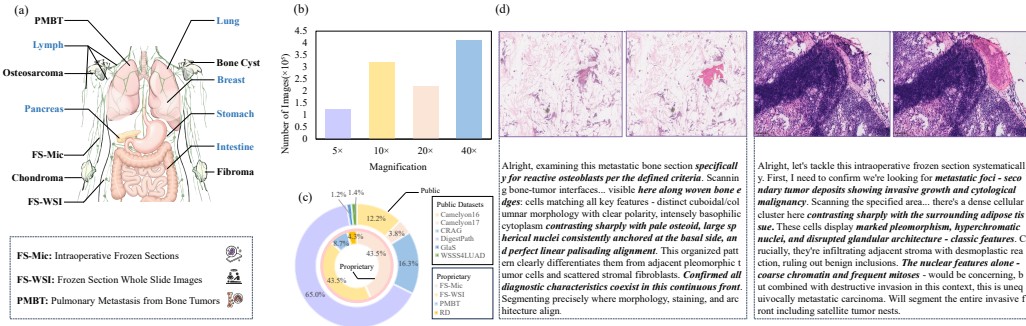

Figure 2: PathChat-SegR1 benchmark statistics and examples. (a) Anatomical distribution showing coverage, with both public and proprietary sources mapped to their corresponding anatomical origins. (b) Magnification-level distribution. (c) Dataset composition. (d) Representative intraoperative frozen section examples with associated reasoning chains.

conditioned on visual features from RuiPath. This stage freezes the LLM backbone while training the RuiPath encoder and intermediate projection layer to establish vision-language alignment.

### 3.3 SUPERVISED FINE-TUNING

After pre-training, supervised fine-tuning teaches PathChat-SegR1 to generate reasoning chains containing <SEG> token while establishing the alignment between language-based reasoning from the VLM and pixel-wise mask generation from the segmentation decoder. To preserve pre-trained knowledge from both pathology and language domains, we adopt parameter-efficient fine-tuning strategies. For the LLM backbone, and RuiPath encoder in VLM, we apply LoRA (Hu et al., 2021) to the self-attention blocks. For the MedSAM encoder, we insert lightweight adapters after the multi-head attention layer and parallel to the MLP layer in each ViT block (Chen et al., 2022), which allow MedSAM encoder to adjust its spatial features. On the contrary, the mask decoder and the adapter that bridges the <SEG> token to the decoder undergo full parameter training, because these components are initialized randomly. The supervised fine-tuning objective jointly optimizes reasoning chain generation and mask prediction, depicted as:

$$\mathcal{L}_{\text{SFT}} = \lambda_{\text{CoT}} \cdot \mathcal{L}_{\text{CE}} + \lambda_{\text{seg}} \cdot (\mathcal{L}_{\text{Dice}} + \mathcal{L}_{\text{BCE}}) \tag{2}$$

where the first term represents an auto-regressive cross-entropy loss for reasoning chain generation. The second term combines the Dice loss $\mathcal{L}_{\text{Dice}}$ and the binary cross-entropy loss $\mathcal{L}_{\text{BCE}}$ for the segmentation mask. Parameters $\lambda_{\text{CoT}}$ and $\lambda_{\text{seg}}$ balance these two terms.

### 3.4 REINFORCEMENT LEARNING WITH SO-GRPO

Standard GRPO assigns rewards uniformly, preventing the VLM in reasoning segmentation from identifying when sufficient semantic context has accumulated to trigger segmentation. We propose Segmentation-Optimized GRPO (SO-GRPO), which extends GRPO to address three challenges: (1) determining optimal <SEG> token generation timing, (2) enabling gradient flow from segmentation quality to reasoning representations, and (3) controlling generation sparsity.

**<SEG> token assignment via GAE.** To identify when the VLM has accumulated sufficient semantic context for segmentation, we replace trajectory-level advantage estimation with Generalized Advantage Estimation (GAE) (Schulman et al., 2018). Formally, it assigns credit to individual time steps during reasoning generation, formulated as $\hat{A}^{\text{GAE}}(s_t, a_t) = \sum_{l=0}^{\infty} (\gamma\lambda)^l \delta_{t+l}$, where $s_t$ represents the reasoning state at time step $t$ (comprising the text query, visual features, and all previously generated reasoning tokens), $a_t$ denotes the token generation action at that step $\delta_t = r_t + \gamma V(s_{t+1}) - V(s_t)$ is the temporal difference (TD) residual that measures the reward improvement between consecutive reasoning states. The state value function $V(s_t)$ estimates the expected cumulative reward from state $s_t$ onward, quantifying how much the current reasoning context contributes to the final segmentation

quality. The discount factor $\gamma$ determines how much future rewards influence current decisions, while $\lambda$ controls the bias-variance trade-off in advantage estimation. By computing advantages at each reasoning step through exponentially weighted sums of temporal difference residuals, GAE evaluates whether generating the <SEG> token at the current step would yield better segmentation outcomes than continuing reasoning. The VLM learns to generate the <SEG> token when $\hat{A}^{\mathrm{GAE}}(s_t, a_t = \text{<SEG>})$ is maximized, indicating that the accumulated reasoning context contains sufficient semantic information and that further text generation would not improve segmentation quality. Moreover, GAE reduces the gradient variance by a factor of $\frac{1-(\gamma\lambda)^{2T}}{1-(\gamma\lambda)^2}$ compared to Monte Carlo estimation, enabling more stable policy updates at the decisive generation step where the model must decide whether to trigger segmentation or continue reasoning.

**Differentiable segmentation performance reward.** Standard segmentation metrics such as Dice and IoU are discrete and non-differentiable. Therefore, using these metrics as performance reward prevents gradient flow from to the <SEG> token representation. We introduce differentiable approximations through probability softening as the performance reward, namely $R_{\mathrm{soft}} = \frac{2\sum_i p_i g_i + \epsilon}{\sum_i p_i + \sum_i g_i + \epsilon}$, where $p_i = \sigma(M_{\mathrm{pred},i})$ are softened probabilities from the predicted segmentation mask, $g_i$ are ground truth, and $\epsilon < 10^{-7}$ ensures numerical stability. The softening operation converts discrete masks into continuous probability distributions, enabling a differentiable path $R_{\mathrm{soft}} \to M_{\mathrm{pred}} \to h_{\mathrm{SEG}}$ where gradients flow from segmentation performance directly to the <SEG> token representation. During optimization, the gradient $\frac{\partial R_{\mathrm{soft}}}{\partial h_{\mathrm{SEG}}}$ guides $h_{\mathrm{SEG}}$ to encode spatial locations and semantic properties that maximize overlap with ground truth masks.

**Sparsity-aware reward.** Reasoning segmentation model may generate <SEG> tokens excessively or prematurely, degrading both computational efficiency and segmentation performance. We formulate this as an information problem, introducing a sparsity-aware reward that encourages generation only when the reasoning chain provides sufficient segmentation-relevant information:

$$R_{\mathrm{sparse}} = \beta_{\mathrm{sparse}} \cdot \mathbb{I}(s_t \in S_{\mathrm{spatial}}) - \gamma_{\mathrm{sparse}} \cdot \mathbb{I}(s_t \notin S_{\mathrm{spatial}}), \tag{3}$$

where $S_{\mathrm{spatial}}$ denotes states containing spatial-semantic information detected via rule-based patterns (*e.g.*, presence of location descriptors such as "boundary" or "region"), $\mathbb{I}(\cdot)$ represents the indicator function, and $\beta_{\mathrm{sparse}}$ and $\gamma_{\mathrm{sparse}}$ are balancing coefficients. This reward formulation penalizes <SEG> generation at non-spatial states by $-\gamma_{\mathrm{sparse}}$ while encouraging it at spatial states by $+\beta_{\mathrm{sparse}}$. Mathematically, this is equivalent to maximizing mutual information $I(M_{\mathrm{gt}}; a_t = \text{<SEG>}|s_t)$ between token generation and segmentation targets.

**Overall reward formulation.** Finally, the overall reward function comprises segmentation quality ($R_{\mathrm{soft}}$), sparsity control ($R_{\mathrm{sparse}}$), spatial grounding ($R_{\mathrm{spatial}}$), format correctness ($R_{\mathrm{format}}$), and conciseness penalty ($R_{\mathrm{len}}$). The spatial reward $R_{\mathrm{spatial}}$ verifies whether the predicted mask centroid aligns with location keywords extracted from the text query following previous work (Liu et al., 2025b). The format reward enforces structured reasoning with proper special token usage like <SEG> (DeepSeek-AI et al., 2025). The conciseness penalty discourages overly long chain-of-thought sequences via $R_{\mathrm{len}} = -\lambda_{\mathrm{len}} \cdot \max(0, \ln(L - L_0 + 1))$, where $L$ denotes reasoning length and $L_0$ is a grace threshold. The overall reward in SO-GRPO can be formulated as

$$\begin{aligned} \mathcal{J}_{\mathrm{SO\text{-}GRPO}}(\theta) = \mathbb{E}_{\tau \sim \pi_\theta} & \left[ \sum_t \hat{A}^{\mathrm{GAE}}(s_t, a_t) \log \pi_\theta(a_t|s_t) \right] \\ & + \lambda_{\mathrm{soft}} \cdot R_{\mathrm{soft}} + \lambda_{\mathrm{sparse}} \cdot R_{\mathrm{sparse}} \\ & + \lambda_{\mathrm{spatial}} \cdot R_{\mathrm{spatial}} + \lambda_{\mathrm{format}} \cdot R_{\mathrm{format}} \\ & + R_{\mathrm{len}} - \lambda_{\mathrm{KL}} \cdot \mathcal{L}_{\mathrm{KL}} \end{aligned} \tag{4}$$

where $\mathcal{L}_{\mathrm{KL}}$ prevents excessive policy deviation from the supervised initialization, and $\lambda_{\mathrm{soft}}$, $\lambda_{\mathrm{sparse}}$, $\lambda_{\mathrm{spatial}}$, $\lambda_{\mathrm{format}}$, $\lambda_{\mathrm{KL}}$ are the balancing coefficients.

**Convergence guarantees through adaptive scheduling.** To ensure stable optimization during PathChat-SegR1 reinforcement learning, we employ learning rate schedules $\alpha_k = \alpha_0/(1 + \eta k)$ satisfying Robbins-Monro conditions $\sum_k \alpha_k = \infty$ and $\sum_k \alpha_k^2 < \infty$. Combined with KL regularization that constrains policy updates via Lyapunov function $V(\theta) = \mathbb{E}_{s \sim \rho}[\mathrm{KL}(\pi^* \| \pi_\theta)]$, we establish

monotonic improvement $V(\theta_{k+1}) \leq V(\theta_k) - \eta \|\nabla V(\theta_k)\|_2^2$. This inequality guaranties that each policy update decreases the divergence from the optimal policy, with the magnitude of the descent controlled by the gradient norm $\|\nabla V(\theta_k)\|_2^2$, ensuring convergence to a local optimum.

## 3.5 PATHCHAT-SEGR1 BENCHMARK

**Data Statistics.** We introduce the PathChat-SegR1 benchmark comprising 118,667 triplets of pathology image, ground-truth mask, query, and reasoning chain spanning diverse pathology types of different morphologies, magnification levels, and clinical scenarios, as shown in Fig. 2. The benchmark integrates pathology images from six public datasets, namely Camelyon16 (Bejnordi et al., 2017), Camelyon17 (Bándi et al., 2019), CRAG (Zhang, 2022), DigestPath (Da et al., 2022), GlaS (Sirinukunwattana et al., 2017), WSSS4LUAD (Han et al., 2022), covering breast, lymph node, colorectal, digestive system, and lung pathologies, accounting for 65.0% of the total dataset. To evaluate performance under authentic diagnostic conditions, we also contribute 43,847 private intraoperative frozen section pathology images from 493 real surgical procedures where patholo­gists must deliver diagnoses within minutes to guide resection decisions. These frozen sections include preparation artifacts such as ice crystal distortions and tissue folding that occur during rapid cryosectioning, mimicking the challenging conditions pathologists encounter during intraoperative consultations (Wang et al., 2025; 2024a). The PathChat-SegR1 benchmark includes both intraop­erative microscope images and postoperative whole slide images, including rare pathologies such as chondromyxoid fibroma and chondroblastoma that are typically absent from standard training datasets. We also include a pulmonary metastasis from a private bone tumors dataset containing 25,764 images with heterogeneous scanner protocols to evaluate cross-domain and cross-stain-style generalization. The benchmark spans four magnification levels (5×, 10×, 20×, 40×) with 40× magni­fication comprising the largest portion to reflect clinical practice where higher resolution is needed for diagnostic assessment.

**Data Annotation Process.** For the public data, we reuse the ground-truth masks and provide annotation for the reasoning chain and reasoning segmentation query. For the private data, we additionally first annotate the segmentation masks. Reasoning chain annotations articulate diagnostic logic rather than simple category labels, requiring downstream models to explain their segmentation decisions through pathology-relevant terminology. We developed a semi-automated annotation pipeline. Specifically, Gemini-2.5-Pro first analyzes annotated regions to generate morphological descriptions contrasting target areas with surrounding tissue. DeepSeek-R1 (DeepSeek-AI et al., 2025) then transforms these descriptions into structured reasoning chains following pathology diagnostic protocols, and three board-certified pathologists with 8-15 years of experience manually review all generated reasoning chain to verify clinical accuracy and diagnostic soundness, where revisions are made correspondingly. This pipeline ensures that reasoning annotations reflect authentic pathology diagnostic workflows while maintaining scalability for large-scale dataset construction.

## 4 EXPERIMENTS

Table 2: Performance comparison across in-domain public benchmarks and private intraoperative frozen sections measured by Dice coefficient. Closed-set methods are trained separately on each dataset, while reasoning segmentation baselines and PathChat-SegR1 use unified training across all eight benchmarks and tested in a zero-shot manner. FS-Mic: Frozen Sections (Microscope); FS-WSI: Frozen Sections (Whole Slide Image).

| Category | Methods | Cam16 | Cam17 | GlaS | Digest | CRAG | WSSS | FS-Mic | FS-WSI |
|---|---|---|---|---|---|---|---|---|---|
| Closed-Set | MedSAM (Ma et al., 2024) | 0.69 | 0.71 | 0.82 | 0.64 | 0.71 | 0.38 | 0.62 | 0.76 |
| | SAM-Path (Zhang et al., 2023) | 0.64 | 0.67 | 0.78 | 0.55 | 0.77 | 0.72 | 0.54 | 0.73 |
| | nnU-Net (Isensee et al., 2021) | 0.74 | **0.79** | **0.91** | 0.67 | **0.94** | **0.84** | 0.69 | 0.82 |
| | BiomedParse (Zhao et al., 2024) | 0.61 | 0.64 | 0.83 | 0.47 | 0.51 | 0.12 | 0.53 | 0.71 |
| Reasoning Seg. | LISA-7B (Lai et al., 2024) | 0.48 | 0.45 | 0.52 | 0.46 | 0.60 | 0.47 | 0.39 | 0.42 |
| | OVSeg (Liang et al., 2023) | 0.41 | 0.39 | 0.43 | 0.38 | 0.49 | 0.39 | 0.31 | 0.34 |
| | SAM4MLLM (Chen et al., 2024) | 0.47 | 0.51 | 0.62 | 0.57 | 0.64 | 0.52 | 0.51 | 0.56 |
| | Seg-Zero (Liu et al., 2025b) | 0.54 | 0.55 | 0.66 | 0.52 | 0.59 | 0.38 | 0.42 | 0.51 |
| | MMR-7B (Jang et al., 2025) | 0.57 | 0.59 | 0.72 | 0.58 | 0.74 | 0.68 | 0.56 | 0.62 |
| | Ours | **0.76** | 0.78 | 0.87 | **0.74** | 0.92 | 0.78 | **0.74** | **0.84** |

**Implementation Details.** All experiments run on 8 NVIDIA H800 GPUs using PyTorch 2.1 with AdamW optimizer (learning rate $1 \times 10^{-4}$) and cosine annealing schedule. The training process uses batch size 64 with gradient accumulation over 2 steps. LoRA adapters apply rank $r = 16$ with dropout probability 0.1 to the attention layers. The MedSAM encoder operates with patch size 16 and applies 75% masking ratio during stain-invariant self-distillation pretraining. SO-GRPO training balances multiple reward components with $\lambda_{\text{soft}} = 0.3$, $\lambda_{\text{sparse}} = 0.2$, $\lambda_{\text{spatial}} = \lambda_{\text{format}} = 0.1$, and KL divergence penalty $\lambda_{\text{KL}} = 0.01$ to maintain policy stability throughout reinforcement learning. We use an 8:1:1 split between training, validation, and testing sets on the proposed PathChat-SegR1. Zero-shot generalization is evaluated in 39,611 unseen pathological images spanning two scenarios, namely the Pulmonary Metastasis of Bone Tumors (PMBT) dataset with 25,764 images, and Rare Disease (RD) with 13,847 images of atypical pathologies such as chondroblastoma-like osteosarcoma and chondromyxoid fibroma that are absent from the training distribution. We further assess in a one-shot manner by providing a single annotated reference during inference on the RD dataset.

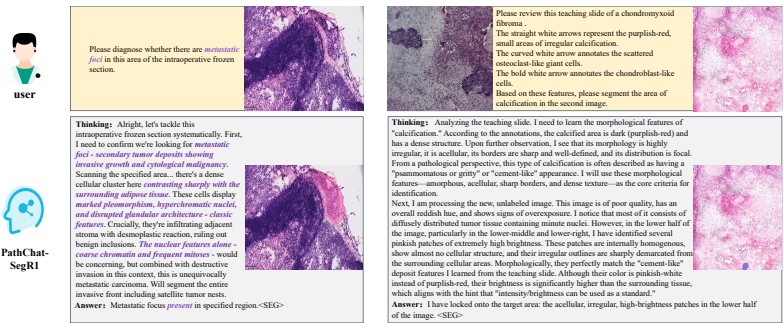

Figure 3: Qualitative comparison of segmentation performance on challenging pathologies in one-shot in-context learning manner. Right panel shows training-free one-shot adaptation on chondromyxoid fibroma, where PathChat-SegR1 analyzes given image and annotation to extract morphological features and applies them to segment calcification in an unlabeled image with different staining appearance.

**In-Domain Performance Evaluation.** Tab. 2 compares PathChat-SegR1 against closed-set and reasoning segmentation methods across eight benchmarks spanning public datasets and clinical frozen sections. PathChat-SegR1 achieves 0.76 Dice on Camelyon16, outperforming the strongest reasoning baseline MMR-7B by 33% relative improvement. In gland segmentation tasks, PathChat-SegR1 reaches 0.87 Dice on GlaS and 0.92 Dice on CRAG in a zero-shot manner, approaching dataset-specific tuned nnU-Net performance while maintaining unified training across all benchmarks. These results demonstrate that pathology-specific visual encoding combined with reasoning capabilities enables strong reasoning segmentation performance of PathChat-SegR1 across diverse tissue types and pathology imaging conditions without requiring task-specific adaptation.

Table 3: Out-of-domain evaluation on unseen pathologies measured by Dice coefficient. PMBT: Pulmonary Metastasis from Bone Tumors; RD: Rare Diseases; RDw/E: Rare Diseases with one-shot example provided during inference.

| Methods | Zero-shot | | One-shot |
| --- | --- | --- | --- |
| | PMBT | RD | RDw/E |
| LISA-7B (Lai et al., 2024) | 0.30 | 0.24 | 0.37 |
| OVSeg (Liang et al., 2023) | 0.29 | 0.25 | 0.36 |
| SAM4MLLM (Chen et al., 2024) | 0.37 | 0.28 | 0.41 |
| Seg-Zero (Liu et al., 2025b) | 0.39 | 0.29 | 0.44 |
| MMR-7B (Jang et al., 2025) | 0.36 | 0.33 | 0.47 |
| **PathChat-SegR1 (Ours)** | **0.58** | **0.53** | **0.72** |

**Out-of-Domain Evaluation on Unseen Pathologies.** Tab. 3 evaluates performance in completely unseen domains under zero-shot and one-shot conditions. On the PMBT dataset containing pulmonary

metastases from bone tumors with heterogeneous staining protocols and scanner variations, PathChat-SegR1 achieves 0.58 Dice under zero-shot evaluation, representing 61% relative improvement over MMR-7B (0.36 Dice). On the RD collection comprising atypical pathologies absent from the training data, including chondroblastoma-like osteosarcoma and chondromyxoid fibroma, PathChat-SegR1 achieves 0.53 Dice compared to MMR-7B's 0.33 Dice. In a one-shot manner in which a single annotated reference is provided during inference, PathChat-SegR1 reaches 0.72 Dice in the RD dataset, showing a 53% improvement over MMR-7B (0.47 Dice). These results indicate that PathChat-SegR1 can be applied to clinical settings where new pathologies/morphologies are encountered.

**One-Shot In-Context Learning for Novel Pathologies.** PathChat-SegR1 enables training-free adaptation through in-context visual learning, where a single annotated reference image provides morphological context for segmenting novel pathologies. Fig. 3 demonstrates this capability on a chondromyxoid fibroma case, where the PathChat-SegR1 receives a teaching pathology image with annotated regions, osteoclast-like giant cells, and chondroblast-like cells. When presented with an unlabeled image exhibiting different staining style, PathChat-SegR1 can identify calcification based on structural features rather than color matching. This one-shot prompting improves RD collection performance from 0.53 Dice under zero-shot conditions to 0.72 Dice, surpassing MMR-7B (0.47 Dice) by 53%. The one-shot prompting capabilities can address the annotation burden inherent to rare pathologies.

Table 4: Component ablation study on the PMBT dataset measured by Dice coefficient. Architecture ablations remove pathology-specific encoders, training ablations remove pretraining stages and supervision objectives, and reward ablations remove SO-GRPO reward components.

| Ablation Type | Component | Dice Score |
|---|---|---|
| – | Full Model | 0.58 |
| Architecture | w/o Ruipath Encoder | 0.42 |
| | w/o Seg-Adapter | 0.52 |
| | w/o Auto-emission | 0.51 |
| Training | w/o MedSAM Encoder Pre-training | 0.51 |
| | w/o SFT | 0.44 |
| | w/o RL | 0.40 |
| Reward Function | w/o Length Reward | 0.56 |
| | w/o Spatial Reward | 0.53 |

**Ablation on Model Architecture and Training Components.** Tab. 4 evaluates component contributions on the PMBT dataset. Removing the RuiPath encoder reduces performance by 0.16 Dice, removing reinforcement learning by 0.18 Dice, and removing SFT by 0.14 Dice. The RuiPath encoder degradation validates that general-domain vision encoders lack pathology-specific knowledge needed to recognize tissue morphologies in out-of-domain samples where morphology differs from natural image distributions. Removing MedSAM encoder pre-training with self distillation reduces performance to 0.51, confirming that stain-invariant self-distillation addresses stain variations. The reinforcement learning ablation demonstrates that SFT alone cannot optimize segmentation token timing, as SFT provides uniform supervision across all reasoning steps rather than learning when semantic context suffices for mask generation.

**Ablation on SO-GRPO Components.** Tab. 5 analyzes SO-GRPO improvements over standard GRPO on the PMBT dataset. Standard GRPO improves performance from 0.40 Dice (SFT only) to 0.53 Dice, while SO-GRPO reaches 0.58 Dice with faster convergence (18K vs 24K training steps). Removing GAE reduces performance to 0.55, confirming that trajectory-level advantage estimation in standard GRPO cannot identify optimal segmentation token timing, whereas GAE assigns credit to individual reasoning steps. Removing differentiable rewards reduces performance to 0.56, as discrete Dice metrics block gradient flow from segmentation quality to token representations.

**Ablation on Reasoning Quality.** Tab. 6 evaluates reasonign chain generation quality on FS-WSI and PMBT datasets measured by BLEU-4 and F1 scores. PathChat-SegR1 achieves 0.315 BLEU-4 and 0.612 F1 on FS-WSI, exceeding reasoning segmentation baselines LISA (0.281, 0.568) and

Table 5: SO-GRPO component ablation on the PMBT dataset. Performance measured by Dice coefficient, convergence measured by training steps to reach optimal performance, and training stability measured by gradient variance during policy updates.

| Configuration | Dice | Steps | Grad. Var. |
|---|---|---|---|
| **Full SO-GRPO** | **0.58** | **18K** | **0.031** |
| *Baseline Methods* | | | |
| SFT only | 0.40 | – | – |
| Standard GRPO | 0.53 | 24K | 0.048 |
| *SO-GRPO Ablations* | | | |
| w/o GAE | 0.55 | 22K | 0.042 |
| w/o Differentiable reward | 0.56 | 20K | 0.038 |
| w/o Sparsity-aware reward | 0.57 | 19K | 0.035 |
| w/o Adaptive scheduling | 0.57 | 21K | 0.033 |

Seg-Zero (0.279, 0.571), as well as medical VLMs LLaVA-Med (0.285, 0.579) and Med-PaLM (0.291, 0.581). The improvement over Med-PaLM, which incorporates medical text pretraining, indicates that pathology-specific visual knowledge from RuiPath contributes to reasoning quality beyond language model capabilities. These results validate that SO-GRPO optimization improves both segmentation accuracy and reasoning interpretability, addressing the requirement for models to articulate diagnostic criteria rather than produce opaque predictions.

Table 6: Reasoning chain quality evaluation on intraoperative frozen sections and rare disease datasets. Quality measured by BLEU-4 for n-gram overlap and F1 for semantic alignment with expert-annotated reasoning chains. FS-WSI: Frozen Sections (Whole Slide Image); PMBT: Pulmonary Metastasis from Bone Tumors.

| Model | FS-WSI | | PMBT | |
|---|---|---|---|---|
| | BLEU-4 | F1 | BLEU-4 | F1 |
| LISA (Lai et al., 2024) | 0.281 | 0.568 | 0.275 | 0.588 |
| Seg-Zero (Liu et al., 2025b) | 0.279 | 0.571 | 0.266 | 0.575 |
| MMR (Jang et al., 2025) | 0.272 | 0.562 | 0.278 | 0.581 |
| LLaVA-Med | 0.285 | 0.579 | 0.288 | 0.595 |
| Med-PaLM | 0.291 | 0.581 | 0.285 | 0.589 |
| **PathChat-SegR1 (Ours)** | **0.315** | **0.612** | **0.311** | **0.607** |

## 5  CONCLUSION

We present PathChat-SegR1, a comprehensive framework addressing the convergent challenges of reasoning segmentation in clinical pathology. Our contributions span three dimensions. First, we develop the PathChat-SegR1 Framework integrating pathology-specific visual encoders with autonomous <SEG> token generation, departing from fixed injection paradigms to enable context-aware segmentation timing. Second, we introduce SO-GRPO, extending standard GRPO through temporal credit assignment, differentiable reward approximation, sparsity control, and convergence guarantees, providing a principled optimization approach for special token generation tasks. evaluation protocols for the field. Third, we construct the PathChat-SegR1 Benchmark comprising 118,667 image-mask-reasoning triplets including challenging intraoperative frozen sections. Our experimental results demonstrate the framework's robustness in complex clinical scenarios. On intraoperative frozen sections with severe artifacts, the model maintains 0.74-0.84 Dice scores. In zero-shot evaluation on rare diseases, it achieves 0.53 Dice, with one-shot adaptation improving to 0.72 without retraining. These results suggest that integrating domain-specific knowledge, reinforcement learning optimization, and structured reasoning can enhance both generalization and interpretability in medical image segmentation. The framework's ability to adapt to unseen pathologies through single-example prompting offers practical value for clinical deployment where annotated data remains scarce.

ACKNOWLEDGMENTS

This work was supported by the National Natural Science Foundation of China (Grant No. 62471288).

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
