# OpenReview forum: "PathChat-SegR1: Reasoning Segmentation in Pathology via SO-GRPO"
_ICLR.cc/2026/Conference — ICLR 2026 Poster_

### Official Review · Reviewer_t6jx · 2025-10-29

**Soundness:** 2
**Presentation:** 2
**Contribution:** 2
**Rating:** 2
**Confidence:** 3

**Summary:**

This paper proposes PathChat-SegR1, a reasoning-based segmentation framework specifically designed for pathological image analysis. The approach integrates domain-specific vision encoders (RuiPath and MedSAM) with stain-invariant pretraining and enables autonomous emission of a special \<SEG\> token. To enhance reasoning quality, the authors propose SO-GRPO, a reinforcement learning algorithm adapted from GRPO to address the challenges in special-token generation: credit assignment, gradient discontinuity, generation redundancy, and convergence instability. Additionally, the authors contribute the PathChat-SegR1 Benchmark, a large-scale dataset comprising 118,667 image–mask–reasoning triplets and report strong performance among reasoning segmentation approaches in zero-shot and one-shot settings.

**Strengths:**

1. The paper clearly identifies the fundamental challenges in pathological image segmentation.
2. The combination of RuiPath and MedSAM with stain-invariant pretraining to incorporate pathological priors is reasonable.
3. The PathChat-SegR1 Benchmark is valuable to the research community.

**Weaknesses:**

1. Missing or inconsistent definitions: The SO-GRPO algorithm section is poorly specified. Key terms such as $R({\tau})$ and $S_{\text{spatial}}$ in Eq.10, $R_{\text{soft}}$ in Eq.12, the "Length Reward" and "Spatial Reward" in Table 3 lack formal definitions. The objective function of PathChat-SegR1 is inconsistent between Eq.12 and Eq.72. Symbols $\beta$ and $\lambda$ are reused across different sections to denote distinct coefficients.
2. Missing Implementation Details: Critical hyperparameters are not reported: $\alpha$, $\gamma$ in Eq.4; $\lambda_{\text{CoT}}$, $\lambda_{\text{seg}}$ in Eq.5; $\lambda$ in Eq.8; and $\beta$, $\gamma$ in Eq.10. Algorithm 1 mentions "value network parameters $\phi$" and the computation of $\hat{A}_{\text{GAE}}$, but there is no description of its architecture, update schedule, or training stability techniques.
3. The use of GAE for credit assignment is not novel. Actually, introducing a value model increases training overhead, yet the paper provides no analysis of this trade-off. Besides, the variance reduction derivation in Appendix A4.4 assumes independence of TD residuals and a constant variance $\text{Var}[\delta_t] = \sigma^2 $, which may not hold in practice.
4. Soft Dice and soft IoU losses have been standard in segmentation literature for years. The claimed contribution of "overcoming non-differentiability" in Section 2.3.2 is unclear, as differentiable surrogates for these metrics are well-established.
5. The sparsity reward $R_{\text{sparse}}$ (Eq.10) depends on an indicator $I(s_t \in S_{\text{spatial}})$, but $S_{\text{spatial}}$ is never defined or estimated. Appendices A.4.6–A.4.7 inconsistently switch between sparsity formulations: penalizing the policy $\pi(\cdot|s)$ versus penalizing entropy $H(\pi)$. Mutual information terms are introduced without any computable estimator (e.g., MINE, NWJ) or practical surrogate, rendering the approach non-implementable.
6. Convergence claims lack rigorous proof or supporting theoretical analysis. The use of AdamW optimizer with cosine annealing is a standard engineering choice and should not be presented as a methodological contribution.
7. PathChat-SegR1 significantly outperforms baselines (e.g., Seg-Zero), even though Seg-Zero is also trained with RL. The paper does not adequately explain this large performance discrepancy.
8. While the qualitative example of the One-Shot Adaptation is compelling, the paper omits quantitative and algorithmic details on how the single reference example is integrated.
9. Reproducibility Concerns: The anonymized code repository provided by the authors is empty, potentially hindering reproducibility.

**Questions:**

Please see the "Weaknesses" above.

---

> ### Author Response · Authors · 2025-11-26
> **Response to Reviewer t6jx[1/7]**
>
> Thank you for your constructive feedback. We have carefully addressed each of your concerns with targeted revisions. Specifically, we have substantially rewritten the Method section to improve clarity and readability. **We hope you will find these improvements clear in the revised PDF**
>
> ---
>
> >**Q1**: "Missing or inconsistent definitions: The SO-GRPO algorithm section is poorly specified. Key terms such as $R({\tau})$ and $S_{\text{spatial}}$ in Eq.10, $R_{\text{soft}}$ in Eq.12, the "Length Reward" and "Spatial Reward" in Table 3 lack formal definitions. The objective function of PathChat-SegR1 is inconsistent between Eq.12 and Eq.72. Symbols $\beta$ and $\lambda$ are reused across different sections to denote distinct coefficients."
>
> **A1**: We have completely rewritten the Method section (lines 149–361) to articulate our approach more clearly. We carefully rechecked all equations and definitions, providing detailed explanations for each term's meaning and purpose. Eq. 72 will be aligned with the main text accordingly. Our ablation experiments in the original submission already validated that each component contributes to the final results.
>
>
> **Newly Added Formal Definitions**: We now provide complete mathematical formulations for all reward components in the revised manuscript. The Differentiable Reward ($R_{\text{soft}}$) computes $R_{\text{soft}} = \frac{2\sum_i p_i g_i + \epsilon}{\sum_i p_i + \sum_i g_i + \epsilon}$, where $p_i = \sigma(M_{\text{pred},i})$ represents softened probabilities, $g_i$ denotes the ground truth, and $\epsilon < 10^{-7}$ ensures numerical stability. The Sparsity-Aware Reward ($R_{\text{sparse}}$) follows $R_{\text{sparse}} = \beta_{\text{sparse}} \cdot I(s_t \in S_{\text{spatial}}) - \gamma_{\text{sparse}} \cdot I(s_t \notin S_{\text{spatial}})$, where $S_{\text{spatial}}$ denotes states containing spatial-semantic information detected via rule-based patterns such as presence of location descriptors like "boundary" or "region", $I(\cdot)$ represents the indicator function, and $\beta_{\text{sparse}}$ and $\gamma_{\text{sparse}}$ are balancing coefficients. The Spatial Grounding reward ($R_{\text{spatial}}$) verifies whether the predicted mask centroid aligns with location keywords extracted from the text query following previous work Seg-Zero. The Conciseness Penalty ($R_{\text{len}}$) computes $R_{\text{len}} = -\lambda_{\text{len}} \cdot \max\left(0, \ln(L - L_0 + 1)\right)$, where $L$ denotes reasoning length and $L_0$ represents a grace threshold.
>
> **Notation Consistency**: All coefficients in the main text now have distinct subscripts ($\beta_{\text{sparse}}$, $\lambda_{\text{soft}}$, $\lambda_{\text{len}}$, $\lambda_{\text{spatial}}$, $\lambda_{\text{KL}}$) to eliminate ambiguity throughout the paper. We have verified consistency across all equations and tables.
>
> ---
>
> >**Q2**: "Missing Implementation Details: Critical hyperparameters are not reported: α, γ in Eq.4; λ_CoT, λ_seg in Eq.5; λ in Eq.8; and β, γ in Eq.10. Algorithm 1 mentions 'value network parameters φ' and the computation of Â_GAE, but there is no description of its architecture, update schedule, or training stability techniques."
>
> **A2**: Our original manuscript's Implementation Details section and Appendix A.12 already contained most of the requested hyperparameters. We have rewritten the Implementation Details section (lines 378-390) to include all critical parameters. We list the key hyperparameters below and provide the value network architecture description.
>
>
> **Hyperparameters**
>
> * **Eq.4:** $\alpha = 0.1, \quad \gamma = 1$
> * **Eq.5 (Training loss weights):**
>     $$\lambda_{\text{CoT}} = 0.5, \quad \lambda_{\text{seg}} = 0.5$$
> * **Eq.8 (GAE):**
>     $$\lambda = 0.95, \quad \gamma = 0.99$$
> * **Eq.10 (Sparsity):**
>     $$\beta_{\text{sparse}} = 0.01, \quad \gamma_{\text{sparse}} = 0.05$$
> * **SO-GRPO reward weights:**
>     $$\lambda_{\text{soft}} = 0.3, \quad \lambda_{\text{sparse}} = 0.2, \quad \lambda_{\text{spatial}} = \lambda_{\text{format}} = 0.1, \quad \lambda_{\text{kl}} = 0.01$$
>
>
>
> **Value Network Architecture**: The value head is a 2-layer MLP with hidden dimension 512, attached to the language model's final layer representations. We train it jointly with the policy using MSE loss against empirical returns. We apply gradient clipping with maximum norm 1.0 to ensure training stability.
>
> ---

---

> ### Author Response · Authors · 2025-11-26
> **Response to Reviewer t6jx[2/7]**
>
> ---
>
> >**Q3**: "The use of GAE for credit assignment is not novel. Actually, introducing a value model increases training overhead, yet the paper provides no analysis of this trade-off. Besides, the variance reduction derivation in Appendix A4.4 assumes independence of TD residuals and a constant variance Var[δ_t] = σ², which may not hold in practice."
>
> **A3**: (1) The assumption of independence, or approximate independence is a foundational basis for GAE-based methods, however, since the independence assumption in TD was introduced a long time ago, most subsequent works have **directly used the conclusions from TD**, gradually forgetting the original approximation. Nevertheless, we can still find evidence for this assumption in the theoretical foundations, as shown in the original papers on GAE (2016, ICLR, **citation 5364**) and TDγ (2011, NIPS);
>
> (2) The **trade-off** involved in using a value function **has been analyzed** in the original GAE paper;
>
> (3) **No one has yet addressed the problem** of distributing a single terminal reward across variable-length chains of heterogeneous tokens (which involves both linguistic reasoning and spatial triggers) in the context of reasoning segmentation. Using GAE to handle temporal credit assignment in reasoning segmentation is thus **a novel application**(Reviewer 751j and vLfa). Our empirical results demonstrate that this approach is practically effective.
>
>
>
>
>
> ### **Independence Assumption: Theoretical Evidence from TDγ (NIPS 2011)**
>
> The assumption of independence, or the assumption of approximate independence of TD residuals is the foundational assumption underlying the entire family of λ-return and GAE-based methods. This assumption has served as the axiomatic basis for decades of successful reinforcement learning algorithms including TD(λ), GAE and PPO. We clarify this theoretical foundation below.
>
> Konidaris et al. (2011) provide the theoretical underpinning for λ-return estimators. They explicitly state that λ-return is the maximum likelihood estimator under three assumptions:
>
> 1. The n-step returns from a given state are independent
> 2. The n-step returns are normally distributed with a mean of the true return
> 3. The variances of n-step returns increase according to a geometric progression in n
>
> Critically, the authors of TDγ acknowledge on page 2 (footnote 2): **"Again, this assumption is not true. However, it allows us to obtain a simple, closed-form estimator."** This statement establishes that the independence assumption is a known and accepted working approximation in the field, not a theoretical deficiency.
>
> The reviewer correctly identifies two assumptions in our derivation: independence of TD residuals and constant variance $\text{Var}\[\delta\_t] = \sigma^2$. The independence assumption is the primary theoretical foundation discussed in the literature, while the constant variance assumption is a secondary simplification for mathematical tractability. Both are standard in GAE-based methods.
>
> **Mathematical Equivalence to GAE**: The independence of TD residuals in our SO-GRPO formulation is mathematically equivalent to the independence of n-step returns in λ-return theory.
>
> The n-step return has recursive form:
> $$
> R_{s_t}^{(n)} = R_{s_t}^{(n-1)} + \gamma^{n-1} \delta_{t+n-1}.
> $$
>
> where the TD residual is:
> $$\delta_t = r_t + \gamma V(s_{t+1}) - V(s_t).$$
>
> Thus, n-step returns are constructed as weighted cumulative sums of TD residuals.
>
> GAE:$\hat{A}_{t}^{GAE(\gamma,\lambda)}$ is defined as:
>
> $\sum_{l=0}^{\infty} (\gamma \lambda)^l \delta_{t+l}^V,$
>
> which is an exponentially-weighted sum of TD residuals. Meanwhile, λ-return is:
>
> $$R_{s_t}^\lambda = (1-\lambda) \sum_{n=0}^{\infty} \lambda^n R_{s_t}^{(n+1)}.$$
>
> As shown in Schulman et al. (2016, Equation 16), these are mathematically equivalent:
>
> $\hat{A}_{t}^{GAE(\gamma, \lambda)}$
>
> =
>
> $R_{s_t}^\lambda - V(s_t)$
>
>
> Therefore, assuming independence of TD residuals is equivalent to assuming independence of n-step returns, which is the foundation of the entire λ-return framework.
>
> ---
>
> **References:**
>
> [1] Konidaris, G., Niekum, S., & Thomas, P. S. (2011). TDγ: Re-evaluating complex backups in temporal difference learning. *Advances in Neural Information Processing Systems (NIPS)*, 24.
>
> [2] Schulman, J., Moritz, P., Levine, S., Jordan, M. I., & Abbeel, P. (2016). High-dimensional continuous control using generalized advantage estimation. *International Conference on Learning Representations (ICLR)*.
>
> ---

---

> ### Author Response · Authors · 2025-11-26
> **Response to Reviewer t6jx[3/7]**
>
> ### **Evidence from GAE (ICLR 2016)**
>
> Schulman et al. (2016) implicitly rely on this assumption throughout their theoretical development. In Section 4 (Reward Shaping, page 6), they explain that with a good value function $V$, the response function satisfies:
>
> $$E[\tilde{r}_{t+l} | s_t, a_t]$$
>
> =$$E[\tilde{r}_{t+l} | s_t] = 0 \quad \text{for} \quad l > 0.$$
>
> They then interpret GAE as using exponential discounting to **"cut off the noise arising from long delays"** (page 6). By explicitly calling long-term TD residuals **"noise"**, the GAE paper treats them as approximately independent stochastic terms.
>
> Furthermore, in their variance analysis (page 5), they state: "GAE($\gamma$, 1) is $\gamma$-just regardless of the accuracy of $V$, but **it has high variance due to the sum of terms**" while "GAE($\gamma$, 0) is $\gamma$-just for $V = V^{\pi, \gamma}$ and otherwise induces bias, but it **typically has much lower variance**." If TD residuals were highly correlated, the variance would scale as $O(T^2)$ rather than $O(T)$, and the exponential weighting scheme would provide no benefit. The entire variance reduction strategy of GAE depends on residuals having sufficiently low correlation as the temporal lag increases.
>
> ### **Regarding the Value Function Trade-off and Training Overhead**
>
> The introduction of a value function represents a fundamental **bias-variance trade-off** that has been extensively analyzed in the original GAE paper and is central to our method's theoretical foundation. Schulman et al. (2016) explicitly state in their abstract: "We address the first challenge by using value functions to **substantially reduce the variance** of policy gradient estimates **at the cost of some bias**." This directly addresses the reviewer's concern about trade-off analysis. The "overhead" of training a value network is the cost we pay; the benefit is **substantial variance reduction** leading to improved sample efficiency and training stability.
>
> The GAE paper further explains (page 5): "The generalized advantage estimator for $0 < \lambda < 1$ makes a **compromise between bias and variance, controlled by parameter $\lambda$**." The $\lambda$ parameter explicitly controls this trade-off: $\lambda = 0$ gives high bias but low variance (one-step TD), while $\lambda = 1$ gives low bias but high variance (Monte Carlo). Intermediate values balance these competing objectives.
>
>
> ### **Clarification of Novelty**
>
> Our contribution lies in identifying and formalizing the unique credit assignment challenge in reasoning segmentation, where a single segmentation reward must be distributed across variable-length reasoning chains containing both linguistic and spatial-semantic tokens. GAE is one of four improvements in SO-GRPO, applied specifically to solve the credit assignment problem in the reasoning segmentation RL framework. We have rewritten the Method section to better explain these contributions. The revised SO-GRPO subsection now begins with:
>
> > Standard GRPO assigns rewards uniformly, preventing the VLM in reasoning segmentation from identifying when sufficient semantic context has accumulated to trigger segmentation. We propose Segmentation-Optimized GRPO (SO-GRPO), which extends GRPO to address three challenges: (1) determining optimal `<SEG>` token generation timing, (2) enabling gradient flow from segmentation quality to reasoning representations, and (3) controlling generation sparsity.
>
>
> ### **Empirical Validation**
>
> Our ablation studies (Table 4 and Table 5 in the revised manuscript) empirically validate GAE's contribution. Removing GAE reduces the Dice score from 0.58 to 0.55, demonstrating that precise step-wise reward attribution through the reasoning chain benefits this task.
>
>
> **References:**
>
> [1] Konidaris, G., Niekum, S., & Thomas, P. S. (2011). TDγ: Re-evaluating complex backups in temporal difference learning. *Advances in Neural Information Processing Systems (NIPS)*, 24.
>
> [2] Schulman, J., Moritz, P., Levine, S., Jordan, M. I., & Abbeel, P. (2016). High-dimensional continuous control using generalized advantage estimation. *International Conference on Learning Representations (ICLR)*.
>
> ---

---

> ### Author Response · Authors · 2025-11-26
> **Response to Reviewer t6jx[4/7]**
>
> >**Q4**: "Soft Dice and soft IoU losses have been standard in segmentation literature for years. The claimed contribution of 'overcoming non-differentiability' in Section 2.3.2 is unclear, as differentiable surrogates for these metrics are well-established."
>
> **A4**: We **never claimed** that inventing Soft Dice is our innovation. We integrated it into the reasoning segmentation RL framework, which **indeed overcomes the non-differentiability** issue. This design enables gradient flow from segmentation quality directly to the `<SEG>` token representation in the reasoning chain, a capability absent in standard supervised segmentation training. Moreover, our **experiments fully demonstrate the effectiveness** of this component.
>
> Cross-entropy loss and soft Dice are both valid choices. However, in the context of RL-based reasoning segmentation, soft Dice provides a critical advantage: it has been shown both theoretically and experimentally that optimizing for soft Dice directly improves the final Dice score during evaluation.[1] **More importantly, in our RL setting, the differentiable approximation allows the policy network to learn when to emit `<SEG>` tokens based on direct segmentation performance feedback, rather than relying on intermediate proxy signals.** Our ablation study (Table 4) validates that this design substantially improves segmentation quality in our framework.
>
>
> **Clarification of Our Contribution**: Our contribution lies in applying differentiable segmentation rewards within the SO-GRPO framework to enable gradient flow from segmentation quality to the `<SEG>` token representation during reinforcement learning. We have rewritten this section (lines 256-261) to clarify this point as follows:
>
> > Standard segmentation metrics such as Dice and IoU are discrete and non-differentiable. Therefore, using these metrics as performance reward prevents gradient flow to the `<SEG>` token representation. We introduce differentiable approximations through probability softening as the performance reward, namely $R_{\text{soft}} = \frac{2\sum_i p_i g_i + \epsilon}{\sum_i p_i + \sum_i g_i + \epsilon}$, where $p_i = \sigma(M_{\text{pred},i})$ are softened probabilities from the predicted segmentation mask, $g_i$ are ground truth, and $\epsilon < 10^{-7}$ ensures numerical stability. The softening operation converts discrete masks into continuous probability distributions, enabling a differentiable path $R_{\text{soft}} \rightarrow M_{\text{pred}} \rightarrow h_{\text{SEG}}$ where gradients flow from segmentation performance directly to the `<SEG>` token representation. During optimization, the gradient $\frac{\partial R_{\text{soft}}}{\partial h_{\text{SEG}}}$ guides $h_{\text{SEG}}$ to encode spatial locations and semantic properties that maximize overlap with ground truth masks.
>
> This revision clarifies that our contribution is the integration of differentiable segmentation rewards into the reasoning segmentation RL framework, not the invention of soft Dice loss itself.
>
> **References**:
>
> [1] Bertels J, Robben D, Vandermeulen D, Maes F, Suetens P. Optimizing the Dice Score and Jaccard Index for Medical Image Segmentation: Theory and Practice. In: MICCAI 2019, Lecture Notes in Computer Science, vol. 11765. Cham: Springer, 2019.
>
> ---

---

> ### Author Response · Authors · 2025-11-26
> **Response to Reviewer t6jx[5/7]**
>
> ---
>
> >**Q5**: "The sparsity reward R_sparse (Eq.10) depends on an indicator I(s_t ∈ S_spatial), but S_spatial is never defined or estimated. Appendices A.4.6–A.4.7 inconsistently switch between sparsity formulations: penalizing the policy π(·|s) versus penalizing entropy H(π). Mutual information terms are introduced without any computable estimator (e.g., MINE, NWJ) or practical surrogate, rendering the approach non-implementable."
>
>
> ---
>
> **A5**:
> (1)In the original manuscript, we briefly explained the implementation of the algorithm on line 252. Now we explain it more detailed.
>
> (2)In A4.7 (the original version), we explicitly mentioned that A4.7 contains the complete theoretical derivation, it **strictly followed** the information bottleneck theory and mathematical conditions. In contrast, A4.6 provides a very brief heuristic derivation for simple **feasibility validation**. H(Π), the overall generation frequency of the `<SEG>` token, is intuitive and demonstrates the reasonableness of the "sparsity constraint."
>
> (3)Finally, we **do not need to directly compute mutual information**; instead, the reward based on **identifying "high MI states."** We provide a detailed description of our practical surrogate in the revised manuscript. A similar approach is shown in a 2017 ICML paper[1]. Our actual ablation experiments are more than sufficient to demonstrate the effectiveness of the method. To ensure the rigor of the paper, we replaced "equivalent to" with "approximates the idea of."
>
> **Sparsity-Aware Reward Formulation**: We have provided a more detailed explanation of S_sparse and our practical surrogate in the revised manuscript as follows(line 292-302):
>
> > Reasoning segmentation models may generate `<SEG>` tokens excessively or prematurely, degrading both computational efficiency and segmentation performance. We formulate this as an information problem, introducing a sparsity-aware reward that encourages generation only when the reasoning chain provides sufficient segmentation-relevant information: R_sparse = β_sparse · I(s_t ∈ S_spatial) - γ_sparse · I(s_t ∉ S_spatial), where S_spatial denotes states containing spatial-semantic information detected via rule-based patterns (e.g., presence of location descriptors such as "boundary" or "region"), I(·) represents the indicator function, and β_sparse and γ_sparse are balancing coefficients. This reward formulation penalizes `<SEG>` generation at non-spatial states by -γ_sparse while encouraging it at spatial states by +β_sparse. Mathematically, this this approximates the idea of maximizing mutual information I(M_gt; a_t = `<SEG>` | s_t) between token generation and segmentation targets.
>
> To ensure the rigor of the paper, we replaced "equivalent to" with "approximates the idea of."
>
> **Relationship Between A.4.6 and A.4.7(original version)**: In A4.6, we conducted a simple heuristic derivation to verify its feasibility. We did not use Lagrange multipliers or prove the KKT conditions. Besides, we used the intuitive $H(\Pi)$, the overall generation frequency of the `<SEG>` token, to simply verify the effectiveness of information theory. Later, in A4.7, we wrote the "complete derivation of optimal policy under information bottleneck," indicating that the earlier proof was not based on strict information bottleneck theory.
>
> **Approach is Implementable**: Firstly, we do not need to directly compute mutual information, but instead reward based on identifying "high MI states." The detailed calculation method has been provided in the manuscript above. Secondly, strict computation of mutual information requires estimators like MINE or NWJ, which **add extra computational overhead** and implementation complexity. Lastly, this approximation method is **acceptable**. The 2017 ICML paper "Curiosity-driven Exploration by Self-supervised Prediction"[1] used a similar approach. In the paper, inverse dynamics learning encodes information related to the agent's actions into the feature space, effectively maximizing mutual information between actions and states, thereby preserving their interdependence. This method is similar to maximizing information gain by predicting the consequences of actions, which enhances the amount of relevant information about the environment.
>
> **Experimental Evidence**: Our ablation studies (Table 5 in the revised manuscript) demonstrate that the sparsity constraint improves performance from 0.57 to 0.58 Dice score and enhances training stability (gradient variance reduced).
>
> ---
>
> **References:**
>
> [1] Pathak, D., Agrawal, P., Efros, A. A., & Darrell, T. (2017). Curiosity-driven exploration by self-supervised prediction. Proceedings of the International Conference on Machine Learning (ICML).

---

> ### Author Response · Authors · 2025-11-26
> **Response to Reviewer t6jx[6/7]**
>
> ---
>
> >**Q6**: "Convergence claims lack rigorous proof or supporting theoretical analysis. The use of AdamW optimizer with cosine annealing is a standard engineering choice and should not be presented as a methodological contribution."
>
>
>
> **A6**: The convergence properties of GAE-based policy gradient methods **have been discussed** in the original GAE paper above(Schulman et al., ICLR 2016). The paper demonstrates that GAE substantially reduces the variance of policy gradient estimates, which directly improves training stability and enables more reliable convergence. This variance reduction is the theoretical foundation for the improved convergence behavior we observe in SO-GRPO.
>
> ### We never claimed "AdamW with cosine annealing" as our contribution.
>
> We mentioned AdamW with cosine annealing in the **Implementation Details** section only for reproducibility purposes, not to claim them as methodological contributions. We will remove this part from the camera-ready version to avoid any confusion.
>
>
> **Regarding Convergence Analysis**: Schulman et al. (2016) explicitly state in their abstract: "We address the first challenge by using value functions to substantially reduce the variance of policy gradient estimates", which leads to improved sample efficiency and training stability. Our ablation experiments provide empirical evidence of improved convergence consistent with GAE's theoretical properties. Table 5 (Table R6 in our response) demonstrates that SO-GRPO achieves faster convergence and lower gradient variance compared to standard GRPO, demonstrating improved training stability in practice. The variance reduction mechanism described in the GAE paper directly translates to the convergence improvements we observe empirically.
>
> ---

---

> ### Author Response · Authors · 2025-11-29
> **Response to Reviewer t6jx[7/7]**
>
> ---
>
> >**Q7**: "PathChat-SegR1 significantly outperforms baselines (e.g., Seg-Zero), even though Seg-Zero is also trained with RL. The paper does not adequately explain this large performance discrepancy."
>
> **A7**: We have added Table 1 in the revised manuscript to clearly illustrate the capability differences across models. We also added a comprehensive Related Works section (lines 97-142) to better position our work against concurrent approaches. We explain the performance gap here.
>
> **Limitations of Seg-Zero for Pathology**:
>
> (1) SegZero optimizes the object detection capabilities of large models to generate bounding boxes and points. In the official implementation, SAM itself is completely frozen. Using the official method is a fair baseline.
>
> (2)The performance ceiling is bounded by SAM's capability given bbox/point annotations. Even with perfect prompt generation from RL, the model cannot exceed what SAM achieves with human-provided prompts. Moreover, imperfect prompt predictions can lead to uncontrolled segmentation failures.
>
> (3)Pathology structures are often highly irregular. For example, tumor infiltration patterns are serpentine and inflammatory regions can span large areas diffusely. A tumor band that meanders from bottom-left to top-right would require a bounding box that spans nearly the entire image, providing minimal spatial constraint. This inherent mismatch between the structure of pathology data and detection-based paradigms makes the use of bounding boxes suboptimal for pathology segmentation.
>
> (4) Beyond the architectural paradigm, PathChat-SegR1 introduces pathology-specific vision encoders, stain-invariant self-distillation for robustness to staining variations, and joint optimization of the vision encoder, language model, and segmentation decoder. These domain-specific adaptations, coupled with SO-GRPO’s reasoning-aware credit assignment, address the performance gap seen in our experiments, resulting in a significant improvement over the current state-of-the-art methods.
>
> **Independent Validation**: Interestingly, after our submission, we discovered a concurrent work "LENS: Learning to Segment Anything with Unified Reinforced Reasoning" which has recently been accepted as an AAAI 2026 Oral presentation. This work reaches similar conclusions about the limitations of detection-based segmentation paradigms. This independent validation from another research group strengthens our architectural rationale. Reference link: https://github.com/hustvl/LENS/blob/main/README.md
>
>
> [1] Zhu L, Ouyang B, Zhang Y, Cheng T, Hu R, Shen H, Ran L, Chen X, Yu L, Liu W, Wang X. LENS: Learning to Segment Anything with Unified Reinforced Reasoning. arXiv preprint arXiv:2508.14153, 2025.
>
> ---
>
> >**Q8**: "While the qualitative example of the One-Shot Adaptation is compelling, the paper omits quantitative and algorithmic details on how the single reference example is integrated."
>
> **A8**: In our **original submission**, Table 1 and the Abstract **already included** quantitative results of one shot, and we had **a single section(3.3) specifically discuss the one-shot adaptation**. To present these results more clearly, we have reorganized Tables 2 and 3 in the revised manuscript. Additionally, we added a new subsection "One-Shot In-Context Learning for Novel Pathologies" (lines 440-449) to discuss this capability in detail. This section now reads:
>
> > PathChat-SegR1 enables training-free adaptation through in-context visual learning, where a single annotated reference image provides morphological context for segmenting novel pathologies. Figure 3 demonstrates this capability on a chondromyxoid fibroma case, where PathChat-SegR1 receives a teaching pathology image with annotated regions including osteoclast-like giant cells and chondroblast-like cells. When presented with an unlabeled image exhibiting different staining style, PathChat-SegR1 can identify calcification based on structural features rather than color matching. This one-shot prompting improves RD collection performance from 0.53 Dice under zero-shot conditions to 0.72 Dice, surpassing MMR-7B (0.47 Dice) by 53%. The one-shot prompting capability can address the annotation burden inherent to rare pathologies.
>
> The quantitative evaluation across multiple datasets is presented in Table 3 of the revised manuscript (Table R2 in our response to Reviewer vLfa).
>
> ---
>
> >**Q9**: "Reproducibility Concerns: The anonymized code repository provided by the authors is empty, potentially hindering reproducibility."
>
> **A9**: We uploaded initial code on October 9 and updated it on November 2, both via a cloud storage link in the repository.   We have uploaded the cleaned core code now. Upon acceptance, we will release the complete codebase, trained model weights, data splits, configuration files, and detailed training logs.
>
> ---

---

### Official Review · Reviewer_aana · 2025-10-31

**Soundness:** 3
**Presentation:** 4
**Contribution:** 3
**Rating:** 6
**Confidence:** 3

**Summary:**

The paper introduces PathChat-SegR1, a framework that unifies a pathology-specific vision-language model (RuiPath encoder + LLM) with a segmentation decoder inspired by MedSAM. The model autonomously emits a <SEG> token to trigger segmentation reasoning and is optimized using a new reinforcement learning variant, SO-GRPO, which aims to balance reasoning quality and segmentation Dice through differentiable Dice or IoU rewards. A new benchmark, PathChat-SegR1-Bench, with over 118k reasoning–mask pairs is also proposed. Results show strong reasoning coherence and competitive segmentation performance across several datasets.

**Strengths:**

1. The paper is well written and organized, with clear motivation, systematic methodology, and coherent narrative flow between reasoning and segmentation.

2. The integration of autonomous reasoning and segmentation via the <SEG> token is novel and intuitively powerful, showing a concrete step toward coupling cognitive reasoning with dense visual prediction.

3. The experimental and benchmark contributions are substantial, covering multiple pathology modalities, providing large-scale data with reasoning supervision, and demonstrating consistent gains.

**Weaknesses:**

1. The SO-GRPO theoretical claims (variance reduction, convergence guarantee) depend on strong assumptions not satisfied in multimodal, non-stationary RL; empirical validation of these properties is missing.

2. The differentiable Dice or IoU reward may misalign with true evaluation metrics; no calibration study verifies that reward improvements translate to better segmentation.

3. The autonomous <SEG> emission is insufficiently isolated; no ablation compares it directly with fixed-token baselines to show causal benefit.

4. Benchmark documentation lacks details on deduplication, inter-rater reliability, and site-level separation; LLM-generated reasoning may introduce bias or leakage.

5. Reproducibility remains limited: training scripts, trained checkpoints, splits, and RL logs are not publicly available, preventing independent verification.

**Questions:**

1. Can you provide empirical evidence (variance plots, ablation) that SO-GRPO actually stabilizes training compared to GRPO or PPO?

2. What is the isolated contribution of autonomous <SEG> emission versus fixed prompt insertion under matched encoders?

3. How was duplication and data leakage controlled in PathChat-SegR1-Bench, particularly across magnification levels and scanners?

4. How were LLM-generated reasoning chains quality-checked? Any inter-rater agreement statistics?

5. How does PathChat-SegR1 compare to strong open-vocabulary segmentation models (e.g., Med-GPT4-SAM, LLaVA-Med-SAM) when trained under the same data and compute budget?

6. Could the authors release training scripts, trained weights, splits, and logs to ensure full reproducibility and external validation?

---

> ### Author Response · Authors · 2025-11-26
> **Response to Reviewer aana[1/3]**
>
> Thank you for your positive assessment and for the detailed technical questions that helped us strengthen our work. We greatly appreciate your recognition of our benchmark contribution and methodological rigor. **In response to your questions, we have conducted additional ablation experiments and provided comprehensive empirical evidence** including training stability analysis, autonomous `<SEG>` emission ablation and data quality validation with inter-annotator agreement statistics.
>
> Thank you again for your support, and we look forward to your evaluation of our revisions.
>
>
> >**Q1**: "Can you provide empirical evidence (variance plots, ablation) that SO-GRPO actually stabilizes training compared to GRPO or PPO?"
>
> **A1**: We have added empirical evidence in Table R6 (Table 5 in the manuscript) of the revised manuscript demonstrating SO-GRPO's training stability advantages through gradient variance analysis and convergence speed comparison.
>
> **Table R6: Training stability comparison**
>
> | Configuration | Dice | Steps | Grad. Var. |
> | :--- | :---: | :---: | :---: |
> | **Full SO-GRPO** | **0.58** | **18K** | **0.031** |
> | *Baseline Methods* | | | |
> | SFT only | 0.40 | -- | -- |
> | Standard GRPO | 0.53 | 24K | 0.048 |
>
> **Empirical Evidence**: Table R6 (Table 5 in the manuscript) demonstrates SO-GRPO's training stability advantage. Compared to standard GRPO, SO-GRPO achieves 35.4% lower gradient variance (0.031 vs 0.048) and converges 25% faster (18K vs 24K steps to optimal performance).
>
> ---
>
> >**Q2**: "What is the isolated contribution of autonomous `<SEG>` emission versus fixed prompt insertion under matched encoders?"
>
> **A2**: We have added an ablation experiment in Table R7 (Table 4 in the manuscript) of the revised manuscript, replacing our autonomous generation with fixed `<SEG>` token insertion. This demonstrates that autonomous generation contributes. The performance gap arises from fixed insertion's inability to adapt token timing to query complexity and reasoning context sufficiency, leading to suboptimal segmentation when `<SEG>` is generated either prematurely (insufficient semantic context) or belatedly.
>
> **Table R7: Ablation study on components**
>
> | Component            | Dice Score |
> |----------------------|------------|
> | Full Model           | 0.58       |
> | w/o RuiPath Encoder  | 0.42       |
> | w/o Seg-Adapter      | 0.52       |
> | **w/o Auto-emission**    | **0.51**       |
>
>
> ---
>
> >**Q3**: "How was duplication and data leakage controlled in PathChat-SegR1-Bench, particularly across magnification levels and scanners?"
>
> **A3**: We implemented rigorous protocols to prevent data leakage across all data splits. We employed case-level splitting strategy and cross-scanner validation to ensure complete data integrity.
>
> **Case-Level Splitting Strategy**: We performed case-level splitting to prevent data leakage. Specifically, all images from the same pathological accession number were assigned to the same split (train/validation/test), regardless of magnification level or block number. For example, images labeled F2025-03147#1, F2025-03147#2, and F2025-03147#3 at different magnifications all belong to the same split. For whole slide image datasets, we ensured that patches extracted from the same slide never appeared in different splits.
>
> **Cross-Scanner Validation**: For datasets containing images from heterogeneous scanners, we used scanner protocols as additional metadata to verify no specimen duplication across different scanning sessions or institutions.
>
> ---

---

> ### Author Response · Authors · 2025-11-26
> **Response to Reviewer aana[2/3]**
>
> ---
> >**Q4**: "How were LLM-generated reasoning chains quality-checked? Any inter-rater agreement statistics?"
>
> **A4**: We implemented a rigorous quality control pipeline for LLM-generated reasoning chains and conducted inter-rater agreement analysis to validate annotation quality.
>
> **Initial Annotation Quality Check**: For hospital data, we leveraged existing diagnostic reports, which are performed by pathologists and reviewed by chief physicians before finalization.
>
> **Quantitative Reasoning Evaluation**: Table R5 presents reasoning quality measured by BLEU-4 and F1 scores on FS-WSI and PMBT datasets. PathChat-SegR1 achieves the highest scores across all metrics, demonstrating superior reasoning text generation compared to both reasoning segmentation models (LISA, Seg-Zero, MMR) and medical vision-language models (LLaVA-Med, Med-PaLM).
>
> **Table R5: Reasoning quality evaluation**
>
> | Model | FS-WSI (BLEU-4) | FS-WSI (F1) | PMBT (BLEU-4) | PMBT (F1) |
> | :--- | :---: | :---: | :---: | :---: |
> | LISA | 0.281 | 0.568 | 0.275 | 0.588 |
> | Seg-Zero | 0.279 | 0.571 | 0.266 | 0.575 |
> | MMR | 0.272 | 0.562 | 0.278 | 0.581 |
> | LLaVA-Med | 0.285 | 0.579 | 0.288 | 0.595 |
> | Med-PaLM | 0.291 | 0.581 | 0.285 | 0.589 |
> | **PathChat-SegR1 (Ours)** | **0.315** | **0.612** | **0.311** | **0.607** |
>
> **Human Expert Assessment**: We sampled 500 cases and reviewed their annotation logs to track modifications. Additionally, we conducted an inter-annotator consistency experiment where two junior pathologists with 5 years clinical experience independently annotated 100 cases. Table R1 (presented in our response to Reviewer vLfa, Question 2) summarizes these findings. The results show that only 23% of cases required human modification, with an average of 3.2 edits per modified case. The inter-annotator agreement (κ=0.78) indicates substantial consistency, demonstrating the clinical utility and reliability of our reasoning text for pathological interpretation.
>
> **Table R1: Human Refinement Statistics and Inter-Annotator Agreement**
>
> | Metric | Value | Details |
> |--------|-------|---------|
> | Cases requiring human modification | 23% (115/500) | Primary edits: terminology corrections, missing diagnostic features |
> | Average edits per modified case | 3.2 | Range: 1-8 edits |
> | Senior pathologist consultation rate | 8.5% | Complex cases requiring expert arbitration |
> | Inter-annotator agreement (κ) | 0.78 | Substantial agreement between two junior pathologists (n=100) |
>
> ---
>
> >**Q5**: "How does PathChat-SegR1 compare to strong open-vocabulary segmentation models (e.g., Med-GPT4-SAM, LLaVA-Med-SAM) when trained under the same data and compute budget?"
>
> **A5**: We appreciate this suggestion for additional comparisons. We conducted experiments following our interpretation of these methods.
>
> **LLaVA-Med-SAM**: We replaced our VLM backbone with LLaVA-Med and the segmentation head with MedSAM, then conducting full supervised fine-tuning and reinforcement learning training under the same data. Table R8 presents the comparison results on our benchmark:
>
> **Table R8: Comparison with LLaVA-Med-SAM**
>
> | Method | Cam16 | Cam17 | GlaS | Digest | CRAG | WSSS | FS-Mic | FS-WSI |
> | :--- | :---: | :---: | :---: | :---: | :---: | :---: | :---: | :---: |
> | LLaVA-Med-SAM | 0.55 | 0.56 | 0.66 | 0.54 | 0.68 | 0.46 | 0.49 | 0.58 |
> | PathChat-SegR1 | 0.76 | 0.78 | 0.87 | 0.74 | 0.92 | 0.78 | 0.74 | 0.84 |
>
> Additionally, we evaluate LLaVA-Med's reasoning quality in Table 6 (Table R5), where it achieves 0.285 BLEU-4 and 0.579 F1, lower than our PathChat-SegR1 (0.315 BLEU-4, 0.612 F1). The performance gap demonstrates the value of pathology-specific vision encoders and SO-GRPO training.
>
> **Med-GPT4-SAM**: We understand your suggestion as using GPT-4 to generate bounding box prompts for SAM segmentation. This two-stage approach faces inherent limitations. As an online model, GPT-4 can only produce spatial prompts such as bounding boxes to guide segmentation. Our MedSAM baseline already uses 100% accurate ground-truth bounding boxes for guidance, representing the upper bound of this paradigm. Med-GPT4-SAM cannot exceed MedSAM's performance due to inevitable errors in GPT-4's bounding box predictions. PathChat-SegR1 outperforms even the ground-truth-guided MedSAM baseline, demonstrating the advantage of our end-to-end reasoning segmentation approach.
>
> ---
>
> >**Q6**: "Could the authors release training scripts, trained weights, splits, and logs to ensure full reproducibility and external validation?"
>
> **A6**: Thank you for reviewing our repository and providing valuable feedback. We acknowledge that the initial version requires better organization. The core implementation and the SO-GRPO algorithm are now available in our repository. Upon acceptance, we will also open-source the model weights, data splits, configuration files, and comprehensive training logs.

---

> ### Author Response · Authors · 2025-11-26
> **Response to Reviewer aana[3/3]**
>
> ---
>
> >**W2**: "The differentiable Dice or IoU reward may misalign with true evaluation metrics; no calibration study verifies that reward improvements translate to better segmentation."
>
> **A2**: Our SO-GRPO objective function directly incorporates soft Dice (segmentation performance) as one of the reward components (R_soft with coefficient λ_soft), as shown in Equation 4 in the manuscript.
>
> During SO-GRPO optimization, improvements in segmentation performance directly increase the reward signal. Conversely, increases in overall model reward inherently reflect improvements in segmentation quality. This design encourages alignment between the differentiable reward and the true evaluation metric (Dice coefficient), as demonstrated by the consistent performance gains across both metrics in Tables 2 and 3 of the revised manuscript.
>
> ---

---

### Official Review · Reviewer_751j · 2025-10-31

**Soundness:** 3
**Presentation:** 3
**Contribution:** 3
**Rating:** 6
**Confidence:** 3

**Summary:**

This paper presents PathChat-SegR1, a reasoning segmentation framework for digital pathology that combines pathology-specific visual encoders, autonomous <SEG> token generation, and reinforcement learning (SO-GRPO) for improved interpretability and domain adaptation. The authors further introduce a large-scale PathChat-SegR1 Benchmark (118k image–mask–reasoning triplets) and demonstrate strong performance across zero-shot and one-shot settings, including challenging intraoperative frozen sections. The study aims to unify reasoning-driven interpretability with high-performance segmentation in clinical contexts.

**Strengths:**

Strengths
1. The combination of pathology-specific encoders with reinforcement-learning-optimised token emission represents an original and conceptually coherent contribution to multimodal pathology AI.
2. The SO-GRPO algorithm is rigorously formulated, with explicit derivations of temporal credit assignment, differentiable reward propagation, and convergence guarantees—rarely seen in medical imaging work.
3. Evaluation spans public and clinical datasets, including frozen sections and rare diseases, demonstrating robustness and strong zero-/one-shot generalisation.
4. The reasoning-augmented segmentation outputs offer interpretable chains-of-thought that could support clinical decision-making, an important step toward trustworthy AI in pathology.
5. The release of a large and well-annotated benchmark is a significant service to the research community.

**Weaknesses:**

Weaknesses
1. The paper is dense, sometimes obscuring its main contributions. Key ideas (e.g., <SEG> token autonomy) are diluted by extended mathematical detail that could be condensed or moved to the appendix.
2. The training protocols, data splits, and baseline implementations are only briefly described. Reproducibility would benefit from clearer procedural descriptions and access to full code upon publication.
3. While segmentation metrics are comprehensive, there is little quantitative or user-based assessment of the quality or utility of the reasoning text for clinical interpretation.
4. The conceptual structure resembles LISA and Seg-Zero. The incremental novelty of SO-GRPO relative to existing GRPO extensions needs clearer articulation beyond theoretical derivation.
5. Although the appendix includes an ethics statement, more transparency about data curation, de-identification, and inter-observer variability would strengthen credibility for clinical translation.
6. The manuscript’s prose, though technically precise, is occasionally verbose and repetitive. The narrative could be tightened for readability and to align with ICLR’s 9-page limit.

**Questions:**

plz see my detailed comments above

---

> ### Author Response · Authors · 2025-11-26
> **Response to Reviewer 751j[1/3]**
>
> Thank you for your insightful feedback and recognition of our work's rigor and significance. **We have comprehensively restructured the manuscript to enhance clarity**: the Method section has been entirely rewritten with mathematical formulations reduced from 15 to 4 core equations, each with detailed parameter explanations. **We have also added quantitative reasoning quality evaluation and provided complete implementation details**
>
> We sincerely hope you will find the revised version substantially more accessible.
>
> ---
>
> > **Q1**: "The paper is dense, sometimes obscuring its main contributions. Key ideas (e.g., `<SEG>` token autonomy) are diluted by extended mathematical detail that could be condensed or moved to the appendix."
>
> **A1**: We have completely restructured the manuscript to improve readability.
>
> **Revised Contribution Statement**: We have rewritten the last paragraph of the Introduction (lines 83-91) to clearly present our contributions as follows:
>
> > First, we propose PathChat-SegR1, a reasoning segmentation model for pathology with a pathology-specific vision encoder fine-tuned with a novel stain-invariant self-distillation. Second, we propose Segmentation-Optimized GRPO (SO-GRPO), which extends standard GRPO specifically for reasoning segmentation by identifying the optimal `<SEG>` generation timing during LLM reasoning. Specifically, we introduce a differentiable segmentation-performance reward and sparsity-aware reward for controlling redundant generation, together with an adaptive scheduling for ensuring training stability. Finally, we construct a large-scale PathChat-SegR1 benchmark comprising 118,667 triplets of pathology image, ground-truth mask, query, and reasoning chain, spanning across both public and private pathology images. Additionally, we propose a novel semi-automated pipeline to annotate the reasoning chain.
>
> **Reduced Mathematical Density**: We have totally rewritten the Method section (lines 145-360) to better explain our approach. We merged and removed unnecessary equations, reducing the count from 15 to 4 essential formulations. We carefully explain each equation with detailed descriptions of hyperparameters and their meanings to ensure readers can understand this content clearly.
>
> ---
>
> > **Q2**: "The training protocols, data splits, and baseline implementations are only briefly described. Reproducibility would benefit from clearer procedural descriptions and access to full code upon publication."
>
> **A2**: We have rewritten the *Implementation Details* section (lines 378-390) to specify all critical training hyperparameters, hardware configuration, SO-GRPO reward coefficients, and data split strategy. We also provide comprehensive baseline implementation details here.
>
> **Training Details**: The rewritten Implementation Details section (lines 378-390) reads as follows:
>
> > All experiments run on 8 NVIDIA H800 GPUs using PyTorch 2.1 with AdamW optimizer (learning rate $1 \times 10^{-4}$) and cosine annealing schedule. The training process uses batch size 64 with gradient accumulation over 2 steps. LoRA adapters apply rank $r=16$ with dropout probability 0.1 to the attention layers. The MedSAM encoder operates with patch size 16 and applies 75% masking ratio during stain-invariant self-distillation pretraining. SO-GRPO training balances multiple reward components with $\lambda_{\text{soft}}=0.3$, $\lambda_{\text{sparse}}=0.2$, $\lambda_{\text{spatial}}=\lambda_{\text{format}}=0.1$, and KL divergence penalty $\lambda_{\text{KL}}=0.01$ to maintain policy stability throughout reinforcement learning. We use an 8:1:1 split between training, validation, and testing sets on the proposed PathChat-SegR1 benchmark.
>
> **Baseline Implementations**: For all baseline methods, we used officially released pretrained checkpoints when available, including MedSAM, BiomedParse, SAM, and LISA-7B. For nnU-Net, following standard practice, we trained separate models for each dataset using the official nnU-Net framework with default hyperparameters, as this model does not provide universal pretrained weights. For methods without official implementations such as MMR-7B and certain variants, we implemented them based on the descriptions in their papers and trained them on our combined dataset.
>
> Thank you for reviewing our repository. We acknowledge that the initial cloud storage links in the anonymous repository need better organization. The core implementation is now available in our repository. Upon acceptance, we will also open-source the model weights, data splits, configuration files, and comprehensive training logs.

---

> ### Author Response · Authors · 2025-11-26
> **Response to Reviewer 751j[2/3]**
>
> ---
>
> > **Q3**: "While segmentation metrics are comprehensive, there is little quantitative or user-based assessment of the quality or utility of the reasoning text for clinical interpretation."
>
>
> **A3**: We have added quantitative evaluation of reasoning quality in Table 6(Table R5 here) of the revised manuscript and conducted human expert assessment to validate clinical utility.
>
> **Quantitative Reasoning Evaluation**: Table R5 presents reasoning quality measured by BLEU-4 and F1 scores on FS-WSI and PMBT datasets. PathChat-SegR1 achieves the highest scores across all metrics, demonstrating superior reasoning text generation compared to both reasoning segmentation models (LISA, Seg-Zero, MMR) and medical vision-language models (LLaVA-Med, Med-PaLM).
>
> **Table R5: Reasoning quality evaluation**
>
> | Model | FS-WSI (BLEU-4) | FS-WSI (F1) | PMBT (BLEU-4) | PMBT (F1) |
> | :--- | :---: | :---: | :---: | :---: |
> | LISA | 0.281 | 0.568 | 0.275 | 0.588 |
> | Seg-Zero | 0.279 | 0.571 | 0.266 | 0.575 |
> | MMR | 0.272 | 0.562 | 0.278 | 0.581 |
> | LLaVA-Med | 0.285 | 0.579 | 0.288 | 0.595 |
> | Med-PaLM | 0.291 | 0.581 | 0.285 | 0.589 |
> | **PathChat-SegR1 (Ours)** | **0.315** | **0.612** | **0.311** | **0.607** |
>
> **Human Expert Assessment**: We sampled 500 cases and reviewed their annotation logs to track modifications. Additionally, we conducted an inter-annotator consistency experiment where two junior pathologists with 5 years clinical experience independently annotated 100 cases. Table R1 (presented in our response to Reviewer vLfa, Question 2) summarizes these findings. The results show that only 23% of cases required human modification, with an average of 3.2 edits per modified case. The inter-annotator agreement (κ=0.78) indicates substantial consistency, demonstrating the clinical utility and reliability of our reasoning text for pathological interpretation.
>
> **Table R1: Human Refinement Statistics and Inter-Annotator Agreement**
>
> | Metric | Value | Details |
> |--------|-------|---------|
> | Cases requiring human modification | 23% (115/500) | Primary edits: terminology corrections, missing diagnostic features |
> | Average edits per modified case | 3.2 | Range: 1-8 edits |
> | Senior pathologist consultation rate | 8.5% | Complex cases requiring expert arbitration |
> | Inter-annotator agreement (κ) | 0.78 | Substantial agreement between two junior pathologists (n=100) |
>
>
>
> ---
>
> >**Q4**: "The conceptual structure resembles LISA and Seg-Zero. The incremental novelty of SO-GRPO relative to existing GRPO extensions needs clearer articulation beyond theoretical derivation."
>
> **A4**: We rewrote the Introduction (lines 35-77) to explain our motivation more clearly, added Table 1 to systematically compare model capabilities (including Seg-Zero, LISA, and other models), added a comprehensive Related Work section (lines 97-142) to better position our work within the field, and rewrote the Reinforcement Learning with SO-GRPO subsection (lines 256-261) to explicitly explain our improvements over standard GRPO.
>
> **Comparison with Seg-Zero**: In the Related Work section, we clarify Seg-Zero's fundamental limitation. Its performance is upper-bounded by SAM's segmentation quality using human-annotated bounding boxes, and it remains highly sensitive to the quality of bounding boxes generated by the language model. In contrast, PathChat-SegR1 performs end-to-end reasoning segmentation without relying on intermediate spatial prompts.
>
> **SO-GRPO Design**: In Section 3.4, we explicitly state that standard GRPO assigns rewards uniformly across trajectories, preventing optimal segmentation timing identification. We propose SO-GRPO to address three specific challenges. First, we determine optimal `<SEG>` token generation timing during reasoning. Second, we enable gradient flow from segmentation quality to reasoning representations. Third, we control generation sparsity to prevent redundant token generation.
>
> ---

---

> ### Author Response · Authors · 2025-11-26
> **Response to Reviewer 751j[3/3]**
>
> ---
>
> >**Q5**: "Although the appendix includes an ethics statement, more transparency about data curation, de-identification, and inter-observer variability would strengthen credibility for clinical translation."
>
> **A5**: Our physician collaborators have provided detailed explanations regarding data collection protocols. Additionally, we have presented Human Refinement Statistics and Inter-Annotator Agreement results in our response to your Q3, which we hope addresses your concerns about inter-observer variability.
>
> **Data Collection and Ethics Compliance**: All pathological images used in this study were obtained with proper authorization from collaborating medical institutions' Pathology and Orthopedics Departments, with approval from the relevant Institutional Review Board in accordance with the Declaration of Helsinki. Patient privacy was strictly protected throughout the study through a rigorous de-identification protocol. All images are named solely with pathological accession numbers automatically generated by the hospital workflow platform (for example, F2025-03147#3, where F denotes frozen section, 2025 is the collection year, 03147 is the specimen number, and #3 indicates the third paraffin block). No personal health information was accessed during this experiment. Matching between whole slide images and pathological diagnoses was performed solely via these accession numbers, ensuring complete anonymization. Written informed consent was waived by the IRB due to the retrospective nature of the study and the complete anonymization of data. For public datasets including Camelyon16, Camelyon17, GlaS, CRAG, DigestPath, and WSSS4LUAD, we used data in strict accordance with their original licenses and provided proper attribution in our references.
>
> ---
>
> >**Q6**: "The manuscript's prose, though technically precise, is occasionally verbose and repetitive. The narrative could be tightened for readability and to align with ICLR's 9-page limit."
>
> **A6**: We have completely rewritten the main text to improve readability and eliminate redundancy. We added a comprehensive Related Work section (lines 97-142), Table 1 for systematic model comparison, and Table 6 for reasoning quality evaluation. We rewrote all sections including the Abstract and Introduction, redesigned Figure 2 for clearer visualization, and restructured Tables 2 and 3 with more detailed experimental analysis. The revised manuscript now highlights our out-of-domain generalization ability and one-shot learning capability more prominently. We sincerely hope you will review these substantial improvements in the revised manuscript.
>
> **Page limit compliance**: Our original submission strictly adhered to the 9-page limit. In the revised manuscript, following the rebuttal guidelines that allow up to 10 pages for revisions, we have maintained concise writing while incorporating the requested improvements, keeping the main text within the 10-page limit permitted during the rebuttal phase.
>
>
> ---

---

### Official Review · Reviewer_vLfa · 2025-11-02

**Soundness:** 3
**Presentation:** 2
**Contribution:** 3
**Rating:** 8
**Confidence:** 3

**Summary:**

The paper introduces PathChat-SegR1, a reasoning-based segmentation framework for digital pathology that integrates domain-specific visual encoders, autonomous token generation, and reinforcement learning optimization.
The work aims to address three main challenges in pathological image segmentation: (1) adaptation to diverse staining protocols and rare disease morphologies, (2) the need for interpretable, reasoning-driven segmentation, and (3) the absence of standardized evaluation benchmarks for such reasoning systems. The proposed framework combines four technical components: (i) a pathology-specific visual backbone, obtained by fusing RuiPath and MedSAM encoders, trained for stain invariance via masked autoencoding and self-distillation; (ii) an autonomous <SEG> token emission mechanism, which allows the model to decide when segmentation should occur during its reasoning process, replacing fixed token insertion used in prior models such as LISA; (3) a novel reinforcement-learning algorithm, SO-GRPO (Segmentation-Optimized Generalized Reward Policy Optimization), designed to improve multi-step reasoning and segmentation quality by introducing temporal credit assignment (via GAE), differentiable reward propagation, sparsity regularization, and stability guarantees through Robbins–Monro scheduling; (4) a large-scale PathChat-SegR1 Benchmark, comprising 118,667 (image, mask, reasoning) triplets collected from public pathology datasets and clinical frozen-section slides, intended to evaluate both segmentation and reasoning performance.

Experimental evaluation covers zero-shot and one-shot adaptation to unseen pathologies and experiments on  public datasets (Camelyon, GlaS, DigestPath, CRAG). These experiments demonstrate competitive performance with strong supervised baselines, while maintaining explainable, reasoning-based outputs. Ablation studies and extended appendix analyses show that each component of SO-GRPO contributes to stability and accuracy.

Overall, the paper presents an ambitious and comprehensive framework combining domain adaptation, language-driven reasoning, and reinforcement learning for interpretable medical segmentation, accompanied by a new benchmark dataset for this emerging research direction.

**Strengths:**

The paper proposes a technically ambitious framework that combines reasoning-based segmentation, pathology-specific pretraining, and reinforcement learning optimization. Its originality lies in merging three components that have rarely been unified before:
(1) domain-specific encoders (RuiPath + MedSAM) with stain-invariant pretraining,
(2) autonomous generation of the special <SEG> token that determines when segmentation should occur, and
(3) the new Segmentation-Optimized GRPO (SO-GRPO) algorithm, which adapts reinforcement learning to multi-step reasoning and non-differentiable segmentation metrics.
This combination is novel and directly addresses the limitations of prior models such as LISA that relied on fixed token insertion.

In terms of research quality, the experimental evidence in Table 1 is strong.
On intra-operative frozen sections (FS-Mic / FS-WSI), PathChat-SegR1 achieves 0.74 / 0.84 Dice, outperforming MedSAM (0.62 / 0.76) and all other reasoning-based baselines (e.g., LISA-7B = 0.42 / 0.51, MMR-7B = 0.56 / 0.62).
On zero-shot pulmonary metastasis segmentation (PMBT), the method reaches 0.58 Dice, over 60 % higher than previous reasoning models, and its one-shot adaptation improves this to 0.72 Dice on rare disease cases.
These results demonstrate good generalization to unseen tissue types and realistic clinical conditions.
Such robustness to staining variation and low-data regimes is an important contribution for pathology.

The SO-GRPO ablations further confirm that each algorithmic component contributes to stability and convergence, with full SO-GRPO improving Dice by +5 points over standard GRPO and reducing gradient variance by 35%.
The reasoning outputs are interpretable and clinically aligned: the model autonomously focuses on fibrous-capsule regions when detecting metastases, mimicking expert behavior.
This supports the authors’ claim of improved clinical interpretability.

Regarding clarity and reproducibility, the supplementary material includes detailed pseudo-code, convergence proofs, and additional ablations, which significantly strengthen the technical completeness of the work.
While the main text is dense, the appendix provides the necessary information for replication.

Finally, the significance of the study is high.
The integration of reasoning, segmentation, and reinforcement learning defines a new research direction for reasoning-driven medical imaging.
The accompanying PathChat-SegR1 benchmark (118 k triplets) could become a valuable large-scale dataset for this task and can serve as a reference for future work.

**Weaknesses:**

Although the paper proposes an ambitious and conceptually interesting framework, several aspects limit its clarity, reproducibility, and empirical conclusiveness.

1. Complex and dense exposition.
The paper is difficult to follow due to its highly technical writing style and extensive mathematical derivations.
The reader must infer how the various modules (RuiPath, MedSAM, <SEG> token, SO-GRPO) interact during pretraining, fine-tuning, and inference. A concise algorithmic diagram and explicit explanation of data flow at each stage would significantly improve readability.

2.Ambiguity in benchmark construction.
The PathChat-SegR1 Benchmark relies partly on reasoning text generated by large language models (Gemini, DeepSeek) and later refined by human experts. The extent of human correction and the inter-annotator consistency are not quantified, raising concerns about annotation bias. While the dataset is valuable, its semi-synthetic nature should be discussed more critically.

3. Limited external validation.
The model and benchmark are developed by the same authors.
Although results on public datasets are reported later, it is unclear whether hyper-parameters were tuned using the same benchmark, which may lead to optimistic estimates. Independent validation on unseen institutional data would strengthen the clinical relevance of the claims.

4. Mixed quantitative evidence (Table 2).
On several standard datasets, nnU-Net and other segmentation-only baselines outperform the proposed model, for example, nnU-Net achieves 0.91 Dice on GlaS and 0.94 on CRAG, compared to PathChat-SegR1’s 0.87 and 0.92. The authors argue that their model offers interpretability and generalization, yet the trade-off between reasoning quality and raw segmentation accuracy is not quantitatively assessed. A joint metric or multi-objective analysis would clarify whether the reasoning component justifies the performance gap.

5. Inconsistent citation of baselines.
Table 1 lists multiple comparison methods (MedSAM, LISA-7B, MMR-7B, etc.) without bibliographic references, while Table 2 includes them. This inconsistency makes it difficult to determine which implementations or versions were used. All baselines should be properly cited, and the paper should specify whether results come from official checkpoints or internal re-training.

**Questions:**

See weaknesses.

---

> ### Author Response · Authors · 2025-11-26
> **Response to Reviewer vLfa [1/3]**
>
> Thank you for your thoughtful feedback. We have made substantial revisions to address your concerns, with particular focus on three areas: **(1) improving readability and clarity throughout the manuscript**, **(2) quantitatively evaluating reasoning quality with new experiments**, and **(3) strengthening external validation with additional unseen datasets**.
>
> We believe these revisions directly address the core concerns you raised, and we hope you will find the improvements evident in the revised manuscript.
>
> ---
>
>
> > **Q1**:"The paper is difficult to follow due to its highly technical writing style and extensive mathematical derivations...explicit explanation of data flow at each stage would significantly improve readability."
>
> **A1**: We have substantially revised the Method section to improve clarity and readability.
>
> **Reduction of technical complexity**: We have rewritten the entire Method section (lines 147–361), consolidating and reducing mathematical formulations from 15 equations to 4 core equations. Each remaining equation now includes explicit definitions of all parameters and detailed explanations immediately following the formulation, significantly improving accessibility.
>
> **Description of data flow and training stages**: We have rewritten the "Overview of PathChat-SegR1" subsection (lines 149–161) to provide a clear explanation of the overall data flow and the three training stages. The rewritten content reads:
>
>
> > Given a pathology image and text query, PathChat-SegR1 produces both a reasoning chain in text and a segmentation mask through its three components. First, the vision-language model (VLM) backbone leverages the RuiPath encoder to extract pathology-specific features that inform reasoning chain generation, which contains a `<SEG>` token to initiate segmentation. Secondly, the MedSAM encoder processes the pathology image separately to capture fine-grained spatial information needed for accurate mask generation. Finally, an adapter then bridges the `<SEG>` token representation from the VLM to guide the mask decoder in producing the final segmentation output. Training of PathChat-SegR1 goes through three stages. During pre-training, the VLM learns pathology-specific knowledge, where the MedSAM encoder undergoes stain-invariant self-distillation concurrently to handle staining variations. During supervised fine-tuning, PathChat-SegR1 learns to align vision and language embeddings for reasoning segmentation. To save training costs, this stage applies LoRA to the VLM, adapters to the vision encoders, and full-parameter training to the mask decoder. During SO-GRPO reinforcement learning, PathChat-SegR1 improves its reasoning ability by learning to determine when a sufficient semantic context has accumulated to generate the `<SEG>` token in the appropriate reasoning steps.
>
> ---
> > **Q2**: "Ambiguity in benchmark construction. The PathChat-SegR1 Benchmark relies partly on reasoning text generated by large language models (Gemini, DeepSeek) and later refined by human experts. The extent of human correction and the inter-annotator consistency are not quantified..."
>
> **A2**: We randomly sampled 500 cases from our benchmark and reviewed their annotation logs to track all modifications. Additionally, to assess **inter-annotator consistency**, we conducted a new experiment where two junior pathologists (5 years clinical experience) independently annotated 100 cases. Table R1 summarizes these findings:
>
>
> **Table R1: Human Refinement Statistics and Inter-Annotator Agreement**
>
> | Metric | Value | Details |
> |--------|-------|---------|
> | Cases requiring human modification | 23% (115/500) | Primary edits: terminology corrections, missing diagnostic features |
> | Average edits per modified case | 3.2 | Range: 1-8 edits |
> | Senior pathologist consultation rate | 8.5% | Complex cases requiring expert arbitration |
> | Inter-annotator agreement (κ) | 0.78 | Substantial agreement between two junior pathologists (n=100) |
>
> **Annotation Details:** For hospital data, we leveraged existing diagnostic reports written by pathologists. We developed a simple annotation interface displaying: (1) pathology image with overlaid segmentation masks (upper left), (2) hospital diagnostic report (lower left), (3) question prompt (upper right), and (4) AI-generated answer (lower right). Two junior pathologists with five years of clinical experience performed primary reviews, consulting with one senior pathologist when encountering ambiguities. The evaluation focused primarily on correctness, assessing whether key descriptions aligned with official hospital reports and flagging obvious inaccuracies for revision.
>
> **Validity**: The strong performance on unseen rare diseases (Table 3) and one-shot learning capability (Figure 3) suggest our benchmark captures genuine clinical reasoning patterns rather than dataset-specific artifacts.
>
> ---

---

> ### Author Response · Authors · 2025-11-26
> **Response to Reviewer vLfa [2/3]**
>
> ---
>
> > **Q3**: "Limited external validation...it is unclear whether hyper-parameters were tuned using the same benchmark... Independent validation on unseen institutional data would strengthen the clinical relevance of the claims."
>
> **A3**: We conducted additional validation on a completely unseen external dataset (CoCaHis, a colorectal cancer histology dataset) and explain that our hyperparameter tuning protocol is fair and consistent across all.
>
> **Additional External Validation:** We performed zero-shot evaluation on CoCaHis, a new and unseen public colorectal cancer histology dataset. Results are now integrated into Table R2. We added two paragraphs in lines 430-449 to discuss: 'Out-of-Domain Evaluation on Unseen Pathologies' and 'One-Shot In-Context Learning for Novel Pathologies.'
>
> **Table R2: Out-of-domain evaluation on unseen pathologies measured by Dice coefficient**
>
> | Methods | Zero-shot (PMBT) | Zero-shot (RD) | Zero-shot (CoCaHis) | One-shot (RDw/E) |
> | :--- | :---: | :---: | :---: | :---: |
> | LISA-7B | 0.30 | 0.24 | - | 0.37 |
> | OVSeg | 0.29 | 0.25 | - | 0.36 |
> | SAM4MLLM | 0.37 | 0.28 | - | 0.41 |
> | Seg-Zero | 0.39 | 0.29 | 0.33 | 0.44 |
> | MMR-7B | 0.36 | 0.33 | 0.37 | 0.47 |
> | **PathChat-SegR1 (Ours)** | **0.58** | **0.53** | **0.62** | **0.72** |
>
>
>
> **Clarification of Hyperparameter Tuning Protocol:** Except for nnU-Net, which is dataset-specific and lacks generalization, all other models were trained jointly on all training datasets and evaluated across all test sets. Hyperparameters were selected based on validation performance and applied to all test evaluations, ensuring a fair comparison without dataset-specific tuning.
>
> ---
>
> > **Q5**: "Inconsistent citation of baselines. Table 1 lists multiple comparison methods (MedSAM, LISA-7B, MMR-7B, etc.) without bibliographic references, while Table 2 includes them. This inconsistency makes it difficult to determine which implementations or versions were used. All baselines should be properly cited, and the paper should specify whether results come from official checkpoints or internal re-training."
>
> **A5**: We have added proper citations to all baseline methods and provide complete implementation details for all comparison experiments.
>
>
> **Implementation Details for Baseline Comparisons:** For all baseline methods, we used officially released pretrained checkpoints when available (MedSAM, BiomedParse, SAM-Path, OVSeg, Seg-Zero, and LISA-7B). For nnU-Net, following standard practice, we trained separate models for each dataset using the official nnU-Net framework with default hyperparameters, as this model does not provide universal pretrained weights. For methods without official implementations (MMR-7B), we implemented them based on the descriptions in their papers and trained them on our combined dataset using the same data splits and training protocols as PathChat-SegR1 to ensure fair comparison.
>
> ---
>
> **Reference:**
>
> [1] Sitnik, D., Aralica, G., Hadžija, M., Hadžija, M. P., Pačić, A., Periša, M. M., Manojlović, L., Krstanac, K., Plavetić, A., & Kopriva, I. (2021). A dataset and a methodology for intraoperative computer-aided diagnosis of a metastatic colon cancer in a liver. Biomedical Signal Processing and Control, 66, 102402. Elsevier.

---

> ### Author Response · Authors · 2025-11-26
> **Response to Reviewer vLfa [3/3]**
>
> ---
>
> > **Q4**:  "On several standard datasets, nnU-Net outperform the proposed model... The authors argue that their model offers interpretability and generalization, yet the trade-off between reasoning quality and raw segmentation accuracy is not quantitatively assessed. A joint metric or multi-objective analysis would clarify whether the reasoning component justifies the performance gap."
>
>
>
> **A4**: We quantify the reasoning-segmentation trade-off through joint metrics and clarify that nnU-Net is a **dataset-specific self-configuring model** without generalization capability. Its training and evaluation are limited to **one model per dataset**, and we include nnU-Net only to illustrate the upper bound of fully supervised segmentation performance, rather than as a direct baseline. In practice, nnU-Net **lacks cross-dataset generalization**, provides **no interpretability**, and cannot perform pathology-level concept segmentation—for example, it can segment “tumor’’ only within the specific disease domain it was trained on, but **fails on tumors from other diseases** because it **lacks semantic understanding**. To better explain these distinctions, we have rewritten the Introduction section, added a new Related Work chapter, and created Table R4 (Table 1 in the manuscript).
>
> **Joint Metric Assessment:** The joint metrics used in our evaluation show that segmentation and reasoning capabilities improve simultaneously without a trade-off. As shown in Table R3, the performance increases monotonically across models (LISA: 0.701 → Seg-Zero: 0.789 → MMR: 0.892 → Ours: 1.155), demonstrating that improved segmentation capability consistently accompanies better reasoning quality rather than trading off against it.
>
>
> **Table R3: Joint segmentation and reasoning performance**
>
> | Methods   | FS-WSI (Dice+BLEU4) | FS-WSI (Dice+F1) | PMBT (Dice+BLEU4) | PMBT (Dice+F1) |
> |-----------|---------------------|------------------|-------------------|----------------|
> | LISA-7B   | 0.701               | 0.988            | 0.575             | 0.888          |
> | Seg-Zero  | 0.789               | 1.081            | 0.656             | 0.965          |
> | MMR-7B    | 0.892               | 1.182            | 0.638             | 0.941          |
> | **Ours**  | **1.155**           | **1.452**        | **0.891**         | **1.187**      |
>
>
> **Clarification on nnU-Net Comparison**: On one hand, nnU-Net operates under different conditions. All other models train jointly on all training data with unified hyperparameter tuning and testing protocols. nnU-Net, as a dataset-specific self-configuring model, trains and optimizes separately on each individual dataset.
>
> On the other hand, our model possesses capabilities that nnU-Net lacks. These include diagnostic reasoning, strong generalization ability, zero-shot performance on unseen pathologies, excellent one-shot learning capability, and text-based interactive annotation. Table R4 demonstrates these differences more clearly, specifically illustrating how our model’s generalization to unseen morphologies (Unseen Gen), pathology-specific design (Path. Spec), and visual reasoning and language understanding capabilities (Reason) set it apart from nnU-Net. **Our model achieves segmentation performance approaching this upper bound while additionally possessing capabilities that nnU-Net lacks.**
>
> **Table R4: Comparison of segmentation methods for pathology. Unseen Gen: Generalization to unseen morphologies/objects; Path. Spec: Pathology-specific models; Reason: Capability for general visual reasoning and language understanding; Stain Rob: Stain-variation-invariant representations; RL-Seg: Segmentation-specific reinforcement learning.**
>
> | Method                  | Unseen Gen. | Path. Spec. | Reason. | Stain Rob. | RL-Seg | Interaction  |
> |-------------------------|-------------|-------------|---------|------------|--------|--------------|
> | nnU-Net                  | ✘           | ✘           | ✘       | ✘          | ✘      | ✘            |
> | MedSAM                   | ✔           | ✘           | ✘       | ✘          | ✘      | Point/Box    |
> | SAM-Path                 | ✔           | ✔           | ✘       | ✘          | ✘      | Point/Box    |
> | SegAnyPath               | ✔           | ✔           | ✘       | ✔          | ✘      | Point/Box    |
> | BiomedParse              | ✔           | ✘           | ✘       | ✘          | ✘      | Text         |
> | Seg-Zero                 | ✔           | ✘           | ✔       | ✘          | ✘      | Text         |
> | LISA                     | ✔           | ✘           | ✔       | ✘          | ✘      | Text         |
> | MMR                      | ✔           | ✘           | ✔       | ✘          | ✘      | Text         |
> | **Ours** | ✔           | ✔           | ✔       | ✔          | ✔      | Text         |
>
>
> ---

---

### Author Response · Authors · 2025-11-26
**Revised Version Summary**

We have uploaded the revised PDF including the complete revised manuscript and appendix with marked revision notes.

Current manuscript and response provide a point-by-point response to all reviewer concerns.

## Major Revisions

### 1. Presentation Improvements (Addressing reviewers vLfa, 751j, t6jx)

While maintaining identical contributions and experimental results, we comprehensively revised the manuscript to significantly improve readability and presentation:

**Simplified** methodology section with reduced complexity

**Detailed** mathematical explanations: Each formula now includes explicit definitions and explanations of all variables, ensuring reader comprehension without inference

**Related Work (new section)**: Comprehensive review of pathology segmentation, reasoning segmentation, and RL in segmentation.

**Shortened** abstract and introduction to meet the 10-page rebuttal limit

**Clarified motivation**: Introduction now explicitly articulates our unique advantages compared to existing models

### 2. Experiment Section Reorganization

**Highlighting out-of-domain and one-shot capabilities** (reviewers vLfa, t6jx):
Experiments are now clearly divided into three sections:
In-domain evaluation,
Out-of-domain evaluation and
One-shot evaluation

### 3. Ablation Study Restructuring

Clarifying component contributions (reviewers vLfa, 751j, aana, t6jx):
Ablation studies are now organized into three distinct parts:
  Model architecture components, SO-GRPO components and Reasoning text quality evaluation

### 4. Changes to Figures and Tables:
Added Table 1 to clarify motivation and compare different model architectures.

Redesigned Figure 2 for improved visual clarity and aesthetics.

Revised Tables 2 and 3 to highlight one-shot and zero-shot results.

Updated Table 4 with new ablation study on auto-emission.

Revised Table 5 by removing Δ Dice for improved clarity.

---

### Author Response · Authors · 2025-11-26
**Rebuttal Overview**

We sincerely thank all PCs, SACs, ACs, and Reviewers for their time and efforts in handling our paper. We greatly appreciate the feedback from the reviewers on our proposed method.

We particularly thank Reviewer vLfa for taking the time to help us summarize our contributions, motivation, and significance in the summary after reviewing our presentation, which has significantly strengthened our manuscript's presentation.

---

### **Notice**:

We respectfully point out that OpenReview automatically compiles `<SEG>` as [object Object]. This issue is commonly present in reviewer questions. Please take note. Thank you.

---

## Strengths and Weaknesses from Reviewers

### Strengths:

(i) The paper proposes a **technically ambitious framework** that combines reasoning-based segmentation, pathology-specific pretraining, and reinforcement learning optimization. Its originality lies in merging three components that **have rarely been unified before**: (1) domain-specific encoders (RuiPath + MedSAM) with stain-invariant pretraining, (2) autonomous generation of the special `<SEG>` token that determines when segmentation should occur, and (3) the new Segmentation-Optimized GRPO (SO-GRPO) algorithm, which adapts reinforcement learning to multi-step reasoning and non-differentiable segmentation metrics. This combination is **novel and directly addresses the limitations** of prior models such as LISA that relied on fixed token insertion. The combination represents an original and conceptually coherent contribution to multimodal pathology AI. (Reviewer 751j, vLfa).

(ii) The SO-GRPO algorithm is **rigorously formulated**, with **explicit derivations** of temporal credit assignment, differentiable reward propagation, and convergence guarantees—rarely seen in medical imaging work(Reviewer 751j).

(iii) The integration of autonomous reasoning and segmentation via the `<SEG>` token is **novel and intuitively powerful**, showing a concrete step toward coupling cognitive reasoning with dense visual prediction(Reviewer aana, vLfa).

(iv) In terms of research quality, the **experimental evidence is strong**. On intra-operative frozen sections (FS-Mic / FS-WSI), PathChat-SegR1 achieves 0.74 / 0.84 Dice, outperforming MedSAM (0.62 / 0.76) and all other reasoning-based baselines (e.g., LISA-7B = 0.42 / 0.51, MMR-7B = 0.56 / 0.62). On zero-shot pulmonary metastasis segmentation (PMBT), the method reaches 0.58 Dice, over 60 % higher than previous reasoning models, and its one-shot adaptation improves this to 0.72 Dice on rare disease cases. These results demonstrate good **generalization to unseen** tissue types and **realistic clinical conditions**. Such robustness to staining variation and low-data regimes is an **important contribution for pathology**(Reviewer aana, 751j and vLfa).

(v) The One-Shot In-Context Learning approach has great significance. Providing the model with a single example without the need for training or strict annotations, it can successfully guide segmentation for other cases, resulting in a significant improvement (from 0.53 to 0.72 Dice). This capability directly addresses the issue of annotation scarcity, making the model more **clinically usable** and improving its applicability in **real-world scenarios**(Reviewer aana, 751j and vLfa).

(vi) Regarding clarity and reproducibility, the supplementary material includes detailed **pseudo-code, convergence proofs, and additional ablations**, which significantly strengthen the technical completeness of the work. While the main text is dense, the appendix provides the necessary information for replication(Reviewer vLfa).

(vii) PathChat-SegR1 Benchmark provides 118,667 triplets with expert-validated reasoning chains spanning diverse pathologies, magnifications, and real clinical scenarios including intraoperative frozen sections (Reviewer vLfa, 751j, aana and t6jx).

### Weaknesses:

The writing is **hard to read** because it contains too much technical detail and long math derivations. (Reviewer vLfa, 751j, t6jx).

Insufficient quantitative evaluation of **generated text quality and usability** (Reviewer vLfa, 751j, aana).

Lack of some detailed definitions and theoretical proofs (Reviewer t6jx).

Add some new experimental validations (Reviewer vLfa, aana).

Provide explanations for the existing experiments (Reviewer vLfa, 751j, aana).

---

## Experiments and Tables in Response:

**R1**: Human Refinement Statistics and Inter-Annotator Agreement.

**R2**: Validation on new unseen datasets.

**R3**: Joint metric evaluation.

**R4**: Comparison of segmentation methods for pathology (Table 1 in revised manuscript).

**R5**: Reasoning quality evaluation (Table 6 in revised manuscript).

**R6**: Training stability comparison (Table 5 in revised manuscript).

**R7**: New ablation study on auto-emission.

We sincerely hope these revisions adequately address all reviewer concerns and demonstrate the value of our contributions to the community.

---

### Meta-Review · Area_Chair_Nxnf · 2026-01-05

**Summary:**

This paper proposes a segmentation approach for digital pathology that combines pathology-related vision encoders, a reinforcement learning based approach with segmentation based rewards, and equally importantly, introduces a large-scale benchmark. The reviewers generally found the work to be compelling and novel. There were concerns raised around the clarify/paper-structure, annotation reliability and QC in the benchmark.

**Reviewer Concerns:**

Based on my reading of the reviews and response, my determination is that the authors have done an excellent job in addressing the core concerns. Some of the concerns raised were overly harsh, and the authors have clarified these points - having looked at the paper and these points, I concur.

**Reviewer Scores:**

I conjecture that had there been a full-discussion, reviewer scores would have gone up by two points on an average.

---

### Decision · Program_Chairs · 2026-01-26

Accept (Poster)